# Fortuitous Forgetting in Connectionist Networks

**Hattie Zhou**[*]
Mila, Université de Montréal

**Ankit Vani**
Mila, Université de Montréal

**Hugo Larochelle**
Mila, CIFAR Fellow, Google Research, Brain Team

**Aaron Courville**
Mila, Université de Montréal, CIFAR Fellow

## Abstract

Forgetting is often seen as an unwanted characteristic in both human and machine learning. However, we propose that forgetting can in fact be favorable to learning. We introduce *forget-and-relearn* as a powerful paradigm for shaping the learning trajectories of artificial neural networks. In this process, the forgetting step selectively removes undesirable information from the model, and the relearning step reinforces features that are consistently useful under different conditions. The forget-and-relearn framework unifies many existing iterative training algorithms in the image classification and language emergence literature, and allows us to understand the success of these algorithms in terms of the disproportionate forgetting of undesirable information. We leverage this understanding to improve upon existing algorithms by designing more targeted forgetting operations. Insights from our analysis provide a coherent view on the dynamics of iterative training in neural networks and offer a clear path towards performance improvements.

## 1 Introduction

Forgetting is an inescapable component of human memory. It occurs naturally as neural synapses get removed or altered over time (Wang et al., 2020), and is often thought to be an undesirable characteristic of the human mind. Similarly, in machine learning, the phenomena of catastrophic forgetting is often blamed for certain failure modes in the performance of neural networks (McCloskey & Cohen, 1989; Ratcliff, 1990; French, 1999)[1]. However, substantial evidence in psychology and neuroscience have shown that forgetting and learning have a symbiotic relationship (Bjork & Bjork, 2019; Gravitz, 2019). A well-known example is the "spacing effect", which refers to the observation that long-term recall is enhanced by spacing, rather than massing, repeated study sessions. Bjork & Allen (1970) demonstrated that the key to the spacing effect is the decreased accessibility of information in-between sessions.

In this work, we study a general learning paradigm that we refer to as *forget-and-relearn*, and show that forgetting can also benefit learning in artificial neural networks. To generalize to unseen data, we want our models to capture generalizable concepts rather than purely statistical regularities, but these desirable solutions are a small subset of the solution space and often more difficult to learn naturally (Geirhos et al., 2020). Recently, a number of training algorithms have been proposed to improve generalization by iteratively refining the learned solution. Knowledge evolution (Taha et al., 2021) improves generalization by iteratively reinitializing one part of the network while continuously training the other. Iterative magnitude pruning (Frankle & Carbin, 2019; Frankle et al., 2019) removes weights through an iterative pruning-retraining process, and outperforms unpruned models in certain settings. Hoang et al. (2018) iteratively utilize synthetic machine translation corpus through back-translations of monolingual data. Furlanello et al. (2018); Xie et al. (2020) employ iterative self-distillation to outperform successive teachers. Iterated learning[2] (Kirby, 2001) has been

---

[*]Correspondance at: zhou.hattie@gmail.com.

[1]Our paper title is derived from that of French (1999), *Catastrophic Forgetting in Connectionist Networks*.

[2]We distinguish the use of the term "iterated learning", which is a specific algorithm, with "iterative training", which refers to any training procedure that includes multiple generations of training.

shown to improve compositionality in emergent languages (Ren et al., 2020; Vani et al., 2021) and prevent language drift (Lu et al., 2020). We propose that many existing iterative algorithms are instances of a more general forget-and-relearn process, and that their success can be understood by studying the shared underlying mechanism.

**Forget-and-relearn** is an iterative training paradigm which alternates between a forgetting stage and a relearning stage. At a high level, we define a forgetting stage as any process that results in a decrease in training accuracy. More specifically, let $\mathcal{D} = \{(X_i, Y_i)\}_{i \in [n]}$ be the training dataset. Let $U$ represent uniform noise sampled from $(0, 1)$, which we use as a source of randomness for stochastic functions. Given a neural architecture, let $\mathcal{F}$ be the set of functions computable by a neural network with that architecture, and for any $N \in \mathcal{F}$, let $\text{Acc}(N) = \frac{1}{n} \sum_{i=1}^{n} \mathbb{I}\{N(X_i) = Y_i\}$ be the training accuracy of $N$. Let $N_t$ be a network trained on $\mathcal{D}$, and let $C := E[\text{Acc}(\tilde{N})]$ represent the performance of a randomly initialized classifier on $\mathcal{D}$. We say that $f : \mathcal{F} \times (0, 1) \to \mathcal{F}$ is a *forgetting operation* if two conditions hold: (i) $P(\text{Acc}(f(N_t, U)) < \text{Acc}(N_t) \mid \text{Acc}(N_t) > C) = 1$; (ii) the mutual information $I(f(N_t, U), \mathcal{D})$ is positive. The first criterion ensures that forgetting will allow for relearning due to a decrease in accuracy. The second criterion captures the idea that forgetting equates to a partial removal of information rather than a complete removal. For standard neural networks, it is sufficient to show that training accuracy is lower than the model prior to forgetting, but higher than chance accuracy for the given task.

**The Forget-and-Relearn Hypothesis.** *Given an appropriate forgetting operation, iterative retraining after forgetting will amplify unforgotten features that are consistently useful under different learning conditions induced by the forgetting step. A forgetting operation that favors the preservation of desirable features can thus be used to steer the model towards those desirable characteristics.*

We show that many existing algorithms which have successfully demonstrated improved generalization have a forgetting step that disproportionately affects undesirable information for the given task. Our hypothesis stresses the importance of selective forgetting, and shows a path towards better algorithms. We illustrate the power of this perspective by showing that we can significantly improve upon existing work in their respective settings through the use of more targeted forgetting operations. Our analysis in Section 5 sheds insight on the mechanism through which iterative retraining leads to parameter values with better generalization properties.

## 2 Existing Algorithms as Instances of Forget-and-Relearn

In this section and the rest of the paper, we focus on two settings where iterative training algorithms have been proposed to improve performance: image classification and language emergence. We summarize and recast a number of these iterative training algorithms as instances of forget-and-relearn. We also relate our notion of forgetting to other ones in the literature in Appendix A5.

**Image Classification.** A number of iterative algorithms for image classification uses a reinitialization and retraining scheme, and these reinitialization steps can be seen as a simple form of forgetting. Iterative magnitude pruning (IMP) (Frankle & Carbin, 2019; Frankle et al., 2019) removes weights based on final weight magnitude, and does so iteratively by removing an additional percentage of weights each round. Each round, small magnitude weights are set to zero and frozen, while the remaining weights are rewound to their initial values. Zhou et al. (2019a) show that the masking and weight rewinding[3] procedure still retains information from the previous training round by demonstrating that the new initialization achieves better-than-chance accuracy prior to retraining. Thus, we can view IMP as a forget-and-relearn process with an additional sparsity constraint. RIFLE (Li et al., 2020) shows that transfer learning performance can be improved by periodically reinitializing the final layer during the finetuning stage, which brings meaningful updates to the low-level features in earlier layers. We can view this reinitialization step as a forgetting operation that enables relearning.

Knowledge evolution (KE) (Taha et al., 2021) splits the network's weights into a *fit-hypothesis* and a *reset-hypothesis*, and trains the network in generations in order to improve generalization performance. At the start of each generation of training, weights in the reset-hypothesis are reinitialized,

---

[3]In this work, we use "rewind" to mean resetting weights back to their original initialization. We use "reset" and "reinitialize" interchangeably to indicate a new initialization.

while weights in the fit-hypothesis are retained. Taha et al. (2021) suggest that KE is similar to dropout where neurons learn to be more independent. However, it is hard to translate the dropout intuition to the case of a single subnetwork. Another intuition compares to ResNet, since KE can induce a zero mapping of the reset-hypothesis without reducing capacity. But this zero-mapping always occurs with sufficient training, and it is unclear why it would benefit generalization.

**Language Emergence.** Several existing algorithms in the language emergence subfield can also be seen as forget-and-relearn algorithms. Emergent communication is often studied in the context of a multi-agent referential game called the Lewis game (Lewis, 2008; Foerster et al., 2016; Lazaridou et al., 2016). In this game, a sender agent must communicate information about the attributes of a target object to a receiver agent, who then has to correctly identify the target object. Compositionality is a desirable characteristic for the language that agents use, as it would allow for perfect out-of-distribution generalization to unseen combinations of attributes.

Li & Bowling (2019) hypothesize that compositional languages are easier to teach, and propose an algorithm to exert an "ease-of-teaching" pressure on the sender through periodic resetting of the receiver. We can instead view the receiver resetting as partial forgetting in the multi-agent model. Ren et al. (2020) use iterated learning to improve the compositionality of the emergent language. Their algorithm consists of (i) an interacting phase where agents learn to solve the task jointly, (ii) a transmitting phase where a temporary dataset is generated from the utterances of a sender agent, and (iii) an imitation phase where a newly-initialized sender is pretrained on a temporary dataset of the previous sender's utterances, before starting a new interacting phase, and this process iterates. A crucial component during imitation is to limit the number of training iterations, which creates a learning bottleneck that prevents the new model from fully copying the old model. This restricted transfer of information can be seen as a forgetting operation, and showcases an alternative approach to selective forgetting that removes information that is harder to learn.

## 3 Partial Weight Perturbation as Targeted Forgetting

In this section, we show that common forms of partial weight perturbation disproportionately affect information that can hinder generalization in both image classification and language emergence settings, and can thus be seen as a targeted forgetting operation under the forget-and-relearn hypothesis. We provide full details on the perturbation method studied in Appendix A1.1.

### 3.1 Image Classification

What information is undesirable depends on the task at hand. For the issue of overfitting in image classification, we want to favor features that are simple, general and beneficial for generalization (Valle-Pérez et al., 2019). Several works have shown that easy examples are associated with locally simpler functions and more generalizable features, while difficult examples require more capacity to fit and are often memorized (Baldock et al., 2021; Yosinski et al., 2014; Kalimeris et al., 2019; Arpit et al., 2017). Thus, as a proxy, we define *undesirable information* here as features associated with memorized examples. In order to evaluate whether partial weight perturbation selectively forgets undesirable information, we create two types of example splits for our training data: (i) easy or difficult examples, and (ii) true or mislabeled examples. At each training iteration, we make a copy of the model and perform a forgetting operation on the copy. We then evaluate both models and measure separately the training accuracy on the two groups of examples. We perform these experiments using a two-layer MLP trained on MNIST[4]. We further extend to a 4-layer convolution model and a ResNet18 on CIFAR-10, and to ResNet50 on ImageNet in Appendix A1.2.

For the definition of easy or difficult examples, we follow prior work (Jiang et al., 2020; Baldock et al., 2021) and use the output margin as a measure of example difficulty. The output margin is defined as the difference between the largest and second-largest logits. We label $10\%$ of training examples with the smallest output margin in the current batch as difficult examples. In the noisy labels experiment, we sample $10\%$ of the training data[5] and assign random labels to this group.

---

[4]We use an MLP with hidden layers 300-100-10, trained using SGD with a constant learning rate of 0.1.

[5]To simplify the task, we restrict to using only 10% of the MNIST training set so that the random label examples can be learned quickly.

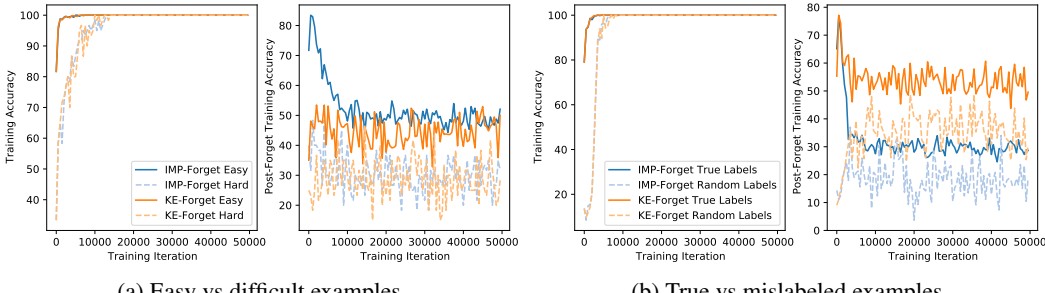

(a) Easy vs difficult examples.

(b) True vs mislabeled examples.

Figure 1: The left panels in (a) and (b) show training accuracy of the two example groups for two types of weight perturbation. The right panels in (a) and (b) show the accuracy of each example group for the same model with weight perturbation applied. Same colors represent the same forgetting operation, and dashed lines represent the accuracy on the difficult or mislabeled examples. Results are averaged over 5 runs.

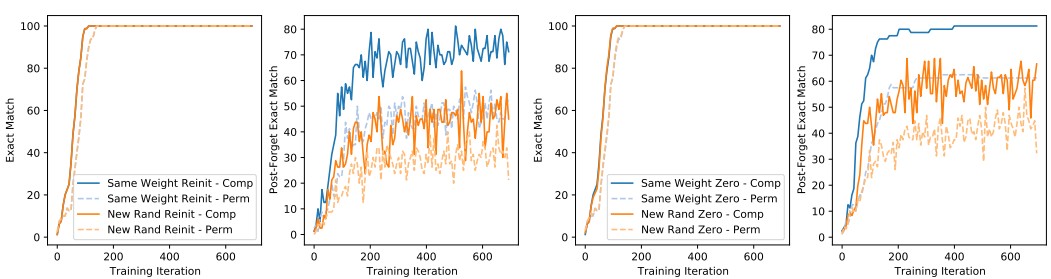

(a) Perturb through reinitialization of masked weights.

(b) Perturb through setting masked weights to 0.

Figure 2: The left panels in (a) and (b) show train accuracy of the compositional (Comp) and non-compositional (Perm) language examples; blue lines here are hidden behind orange lines. The right panels in (a) and (b) show the accuracy of each group for the same model with weight perturbation applied. Same colors represent the same forgetting operation, and dashed lines represent the accuracy on the non-compositional examples. Results are averaged over 5 runs.

We consider KE-style perturbation to be the reinitialization of a random subset of the weights, and IMP-style perturbation to be the zeroing of small magnitude weights and rewinding of the remaining weights. More details are provided in Table A1. Figure 1 shows that both types of weight perturbation adversely impact difficult or mislabeled example groups more severely than the easy or true labeled examples, even when both example groups reach $100\%$ train accuracy. Related to our observations, Hooker et al. (2019) show that pruned models disproportionately affect tails of the data distribution. There are also theoretical works tying robustness-to-parameter-noise to compressibility, flat minima, and generalization (Zhou et al., 2019b; Arora et al., 2018; Foret et al., 2021).

## 3.2 COMPOSITIONAL LANGUAGE EMERGENCE

In the version of the Lewis game we study, the objective is to identify objects with two attributes: SHAPE and COLOR. The sender receives a one-hot representation of the SHAPE and COLOR attributes of an object and produces a discrete message. The receiver takes this message as input and needs to classify the correct object from a set of candidate objects. The two agents are typically parameterized as LSTMs (Hochreiter & Schmidhuber, 1997) and trained simultaneously using REINFORCE (Williams, 1992) or Gumbel-softmax (Jang et al., 2017; Maddison et al., 2017). A compositional language would have a consistent and independent attribute-symbol mapping for all input objects. This is desirable for out-of-distribution generalization, since unseen combinations of attributes will still have a meaningful language output. Thus, the *undesirable information* in this task corresponds to a non-compositional language. We define two candidate languages as designed by Li & Bowling

(2019), and use them to provide possible targets to the model. One language is compositional, where each attribute (4 shapes and 8 colors) is represented by a distinct and consistent symbol-position. The other language is non-compositional, which is formed by randomly permuting the mappings between messages and objects from the compositional language. In this case, each object still has a unique message, but the individual symbols do not correspond to a consistent meaning for shape or color. An example of a compositional and permuted language is provided in Table A2 and A3.

Since the sender controls the language output, we simplify the Lewis game here and train only the sender model to fit a given language through supervised learning. The target language is formed by mixing the two pre-specified languages. For the 32 objects in the dataset, half of them are mapped to the compositional language and half to the non-compositional language. Similar to the experiments in Section 3.1, we make a copy of the model at each training iteration, and perform the forgetting operation on the copy. We evaluate both models and measure the accuracy of the two example groups separately. For the weight perturbations, we consider either a mask based on initial weight magnitude or a random mask. For both mask criteria, we either reinitialize the masked weights or set them to 0. More details can be found in Table A4. As shown in Figure 2, we find that all forms of weight perturbation under study lead to more forgetting of the non-compositional language.

## 4 TARGETED FORGETTING IMPROVES PERFORMANCE

According to the forget-and-relearn hypothesis, the forgetting step should strike a balance between enabling relearning in the next generation and retaining desirable information from the previous generation. We show in this section that through targeted forgetting of undesirable information and preserving of desirable information, we can significantly improve upon existing algorithms.

### 4.1 REDUCING OVERFITTING IN IMAGE CLASSIFICATION

We build upon the KE algorithm since it is widely applicable and leverages a large number of iterative training generations. Following Taha et al. (2021), we study tasks that have a small number of training examples per class, where the network is prone to overfitting, and adopt the same experimental framework unless otherwise specified. Detailed hyperparameters, training procedure, and dataset descriptions can be found in Appendix A2.1-A2.2.

As described by Taha et al. (2021), in KE the mask criteria $M$ is determined prior to training and generated randomly for weight-level splitting (WELS), or based on the location of the kernel for kernel-level splitting (KELS). Given a split-rate $s_r$, $s_r\%$ of weights (for WELS) or kernels (for KELS) in each layer $l$ has a corresponding $M^l$ value of 1. For a neural network parameterized by $\Theta$, we define the *fit-hypothesis* as $M \odot \Theta$ and *reset-hypothesis* as $(1 - M) \odot \Theta$. At the start of each generation, the weights in the fit-hypothesis are kept the same, and the weights in the reset-hypothesis are reinitialized. Then, the network is trained from this new initialization for $e$ epochs, where $e$ is the same for each generation.

Can we perform the forgetting step in a way that more explicitly targets difficult examples in order to improve generalization performance? Baldock et al. (2021) introduce *prediction depth* as a measure of example difficulty in a given model. Prediction depth refers to the layer after which an example's activations can generate the same predictions using a KNN probe as the model's final predictions. They show that increasing prediction depth corresponds to increasing example difficulty. They study this correspondence using various notions of example difficulty, including output margin, adversarial input margin, and speed of learning for each data point.

Based on the observations in Baldock et al. (2021), we hypothesize that by reinitializing the later layers of the neural network, we can remove information associated with difficult examples more precisely than the mask criteria used in KE. Thus, we propose a new forgetting procedure called *later-layer forgetting (LLF)*. Given a layer threshold $L$, we define the LLF mask criterion for each layer $l$ as:

$$M_{\text{LLF}}^l = \left\{ \begin{array}{ll} 1 & \text{if } l < L \\ 0 & \text{if } l \geq L \end{array} \right. \tag{1}$$

We note that the concept of difficult examples is used only as a proxy for information in the network that is more complex, brittle, and specialized. We do not suggest that these examples should not be

learned. In fact, memorization of examples in the tails of the data distribution are often important for generalization (Feldman & Zhang, 2020), which we also achieve in the relearning stage.

Following Taha et al. (2021), we perform LLF on top of non-iterative loss objectives designed to reduce overfitting. We report performance with label-smoothing (SMTH) (Müller et al., 2019; Szegedy et al., 2016) with $\alpha = 0.1$, and the CS-KD regularizer (Yun et al., 2020) with $T = 4$ and $\lambda_{cls} = 3$. Additionally, we add a long baseline which trains the non-iterative method for the same total number of epochs as the corresponding iterative methods. We present results for ResNet18 (He et al., 2016) in Table 1 and DenseNet169 (Huang et al., 2017) in Table A10. We find that LLF substantially outperforms both the long baseline and the KE models across all datasets for both model architectures.

Taha et al. (2021) introduced KE to specifically work in the low data regime. LLF as a direct extension of KE is expected to work in the same regime. For completeness, we also evaluate LLF using ResNet50 on Tiny-ImageNet (Le & Yang, 2015), and WideResNet (Zagoruyko & Komodakis, 2016) and DenseNet-BC on CIFAR-10 and CIFAR-100 (Krizhevsky et al., 2009) in Appendix A2.3. We see a significant gain on Tiny-ImageNet, but not on CIFAR-10 or CIFAR-100.

**Analysis of Related Work.** Surprisingly, we find that when compared to the long baselines with equal computation cost, KE actually *underperforms* in terms of generalization performance, which is contrary to the claims in Taha et al. (2021). We can understand this low performance by looking at the first few training epochs in each generation in Figure 3a. We see that after the first generation, the training accuracy remains near 100% after the forgetting operation. This lack of forgetting suggests that KE would not benefit from retraining under the forget-and-relearn hypothesis. We also observe that KE experiences optimization difficulties post forgetting, as evidenced by a decrease in training accuracy in the first epochs of each generation. This could explain why it under-performs the long baseline. Although KE can enable a reduction in inference cost by using only the slim fit-hypothesis, our analysis reveals that this benefit comes at the cost of lower performance.

Concurrent work (Alabdulmohsin et al., 2021) studies various types of iterative reinitialization approaches and proposes a layer-wise reinitialization scheme, which proceeds from bottom to top and reinitializing one fewer layer each generation. We refer to this method as LW and provide a comparison to their method in Table 1. We find that LLF outperforms LW across all datasets we consider. More discussion on this comparison can be found in Appendix A2.4.

**Analysis of LLF.** In order to view the success of LLF and other similar iterative algorithms in terms of the selective forgetting of undesirable information, we should also observe that selective forgetting of *desirable* information should lead to worse results. We can test this by reversing the LLF procedure: instead of reinitializing the later layers, we can reinitialize the earlier layers. The standard LLF for Flower reinitializes all layers starting from block 4 in ResNet18, which amounts to reinitializing 75% of parameters. The reverse versions starting from block 4 and block 3 reinitializes 25% and 6% of weights respectively. We see in Figure 3b that the reverse experiments indeed perform worse than both LLF and the long baseline.

Additionally, by looking at how prediction depth changes with the long baseline compared to LLF training in Figure A3, we can observe that LLF pushes more examples to be classified in the earlier layers. This further suggests that forget-and-relearn with LLF encourages difficult examples to be relearned using simpler and more general features in the early layers.

## 4.2 INCREASING COMPOSITIONALITY IN EMERGENT LANGUAGES

In order to promote compositionality in the emergent language, Li & Bowling (2019) propose to periodically reinitialize the receiver during training. They motivate this by showing that it is faster for a new receiver to learn a compositional language, and hypothesize that forcing the sender to teach the language to new receivers will encourage the language to be more compositional. As is common in this line of work (Brighton & Kirby, 2006; Lazaridou et al., 2018; Li & Bowling, 2019; Ren et al., 2020), compositionality of the language is measured using topographic similarity ($\rho$), which is defined as the correlation between distances of pairs of objects in the attribute space and the corresponding messages in the message space. Li & Bowling (2019) show that $\rho$, measured using Hamming distances for objects and messages, is higher with the ease-of-teaching approach than training without resetting the receiver.

Table 1: Comparing KE and Later Layer Forgetting for ResNet18. Results are mean accuracy and standard error over 3 runs and reported for hyperparameter settings with best validation performance. KE experiments use KELS split with a split rate of 0.8. LLF uses $L \in \{10, 14\}$, corresponding to block 3 and 4 in ResNet18. N3, N8, N10 indicate the additional number of training generations on top of the baseline model. LLF consistently outperforms all other methods.

| Method | Flower | CUB | Aircraft | MIT | Dog |
|---|---|---|---|---|---|
| Smth (N1) | 51.02 ±0.09 | 58.92 ±0.24 | 57.16 ±0.91 | 56.04 ±0.39 | 63.64 ±0.16 |
| Smth long (N3) | 59.51 ±0.17 | 66.03 ±0.13 | 62.55 ±0.25 | 59.53 ±0.60 | 65.39 ±0.55 |
| Smth + KE (N3) | 57.95 ±0.65 | 63.49 ±0.39 | 60.56 ±0.36 | 58.78 ±0.54 | 64.23 ±0.05 |
| Smth + LLF (N3) (**Ours**) | **63.52** ±0.13 | **70.76** ±0.24 | **68.88** ±0.11 | **63.28** ±0.69 | **67.54** ±0.12 |
| Smth long (N10) | 66.89 ±0.23 | 70.50 ±0.13 | 65.29 ±0.51 | 61.29 ±0.49 | 66.19 ±0.03 |
| Smth + KE (N10) | 63.25 ±0.17 | 66.51 ±0.07 | 63.32 ±0.30 | 59.58 ±0.62 | 63.86 ±0.20 |
| Smth + LLF (N10) (**Ours**) | **70.87** ±0.41 | **72.47** ±0.31 | **70.82** ±0.50 | **64.40** ±0.58 | **68.51** ±0.39 |
| Smth + LW (N8) | 68.43 ±0.27 | 70.87 ±0.15 | 69.10 ±0.27 | 61.67 ±0.32 | 66.97 ±0.24 |
| Smth + LLF (N8) (**Ours**) | **69.48** ±0.24 | **72.30** ±0.28 | **70.37** ±0.49 | **63.58** ±0.16 | **68.45** ±0.25 |
| CS-KD (N1) | 57.57 ±0.61 | 66.61 ±0.02 | 65.18 ±0.68 | 58.61 ±0.25 | 66.48 ±0.12 |
| CS-KD long (N3) | 64.44 ±0.62 | 69.50 ±0.24 | 65.31 ±0.67 | 57.16 ±0.34 | 66.43 ±0.24 |
| CS-KD + KE (N3) | 63.48 ±1.30 | 68.76 ±0.57 | 67.16 ±0.23 | 58.88 ±0.63 | 67.05 ±0.31 |
| CS-KD + LLF (N3) (**Ours**) | **67.20** ±0.51 | **72.58** ±0.02 | **71.65** ±0.21 | **62.41** ±0.45 | **68.77** ±0.24 |
| CS-KD long (N10) | 68.68 ±0.28 | 69.59 ±0.40 | 64.58 ±0.07 | 56.12 ±0.43 | 64.96 ±0.17 |
| CS-KD + KE (N10) | 67.29 ±0.74 | 69.54 ±0.60 | 68.70 ±0.33 | 57.61 ±0.91 | 67.11 ±0.11 |
| CS-KD + LLF (N10) (**Ours**) | **74.68** ±0.19 | **73.51** ±0.35 | **72.01** ±0.23 | **62.89** ±0.59 | **69.20** ±0.12 |
| CS-KD + LW (N8) | **73.72** ±0.74 | 71.81 ±0.21 | 70.82 ±0.34 | 59.18 ±0.41 | 68.09 ±0.24 |
| CS-KD + LLF (N8) (**Ours**) | 73.48 ±0.31 | **73.47** ±0.35 | **71.95** ±0.23 | **62.26** ±0.47 | **69.24** ±0.29 |

We challenge this "ease-of-teaching" interpretation in Li & Bowling (2019) and offer an alternative explanation. Instead of viewing the resetting of the receiver as inducing an ease-of-teaching pressure on the sender, we can view it as an asymmetric form of forgetting in the two-agent system. This form of forgetting is sub-optimal, since we care about the quality of the *messages*, which is controlled by the sender. This motivates a balanced forgetting approach for the Lewis game, whereby both sender and receiver can forget non-compositional aspects of their shared language. Therefore, we propose an explicit forgetting mechanism that removes the asymmetry from the Li & Bowling (2019) setup. We do this by partially reinitializing the weights of both the sender and the receiver and refer to this method as *partial balanced forgetting (PBF)*. We use the "same weight reinit" perturbation method from Section 3.2 with 90% of weights masked. Further training details are found in Appendix A3. We show in Figure 4c that this method, which does not exert an "ease-of-teaching" pressure, significantly outperforms both no resetting and resetting only the receiver in terms of $\rho$.

## 5 UNDERSTANDING THE FORGET-AND-RELEARN DYNAMIC

In Section 4, we discussed the importance of targeted forgetting in the forget-and-relearn dynamic. In this section, we study the mechanism through which iterative retraining shapes learning.

### 5.1 IMAGE CLASSIFICATION

We consider two hypotheses for why iterative retraining with LLF helps generalization. We note that these hypotheses are not contradictory and can both be true.

1. *Later layers* improve during iterative retraining with LLF. Due to co-adaptation during training, the later layers may be learning before the more stable early layer features are fully developed (notably in early epochs of the first generation), which might lead to more overfitting in the later

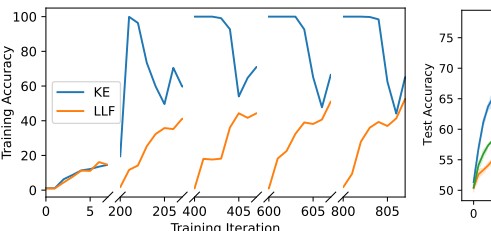

(a) Train accuracy at initialization and in the first 6 epochs of training for 5 generations.

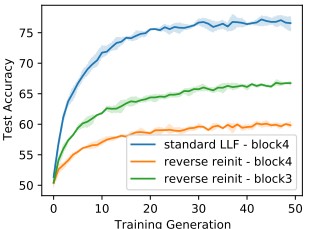

(b) Test accuracy of resetting early layers vs. later layers

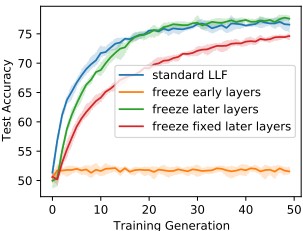

(c) Test accuracy for different freeze layer settings

Figure 3: Analysis experiments performed on Flower dataset with ResNet18. (b)-(c) show the min, max, and average of 3 runs. (a) shows that KE induces no forgetting after the first 2 generations. (b) shows that resetting early layers significantly under-performs LLF. (c) shows that freezing early layers prevents any improvement in subsequent training, while freezing later layers still leads to significant improvements.

layers in some settings. Reinitializing the later layers might give them an opportunity to be relearned by leveraging fully developed features from the earlier layers.

2. *Early layers* improve during iterative retraining with LLF. Since the early layers are continuously refined through iterative training, the features which are consistently useful under new learning conditions may be amplified. These features may be beneficial to generalization.

To test hypothesis 1, we freeze the early layer after the initial training generation and reinitialize the later layers for retraining in subsequent generations. If the later layers improve as a result of learning on top of well-developed early layer features, we should expect a performance improvement during relearning even when early layers are frozen. As shown in Figure 3c under "freeze early layers", we observe no significant performance improvement in this setting, suggesting that simply relearning later layers has little effect on generalization performance.

To test hypothesis 2, we design a reverse experiment where the later layers (except for the output FC layer) are reinitialized and *then* frozen at each subsequent relearning generation. The early layers are trained continuously in each generation. As shown in Figure 3c under "freeze later layers", we find that even with a random frozen layer in the middle, iterative retraining of the early layers still improves performance significantly. To test the interpretation that early layers improve through learning under different conditions, we perform the same experiment as before, except we keep the later layers frozen at the same initialization in each generation. This reduces the amount of variability seen during iterative retraining. As shown in Figure 3c under "freeze fixed later layers", we find the performance of this to be much worse than the version with a different reinitialization each generation, demonstrating the importance of having variable conditions for relearning. These analyses suggest that the main function of iterative retraining is to distill features from the unforgotten knowledge that are consistently useful under different learning conditions.

## 5.2 COMPOSITIONALITY

The analysis in Section 5.1 supports the hypothesis that features that are consistently learned across generations are strengthened in the iterative retraining process. If this hypothesis explains why iterative algorithms improve compositionality in the Lewis game, it would predict that (i) compositional mappings are consistently learned across generations, (ii) the compositional mapping is strengthened during repeated retraining, and (iii) $\rho$ increases during the retraining stage.

For iterated learning, Ren et al. (2020) show that by limiting the number of training iterations in the imitation phase, the new sender can learn a language with better $\rho$ than the previous one, arguing that the compositional components of the language are learned first. The success of iterated learning has primarily been attributed to this increase in compositionality during the imitation phase, which follows the explanation in the cognitive science theories from which iterated learning derives (Kirby, 2001). However, when viewed as an instance of forget-and-relearn, our hypothesis predicts that the

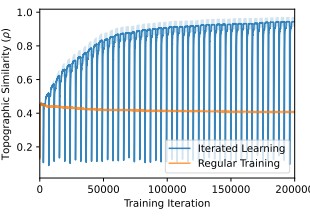
(a) Iterated learning.

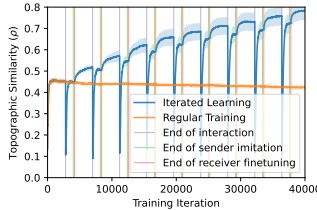
(b) Iterated learning, zoomed to the first 40000 steps.

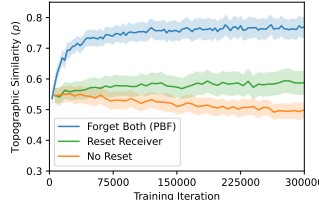
(c) Ease-of-teaching.

Figure 4: Topographic similarity ($\rho$) in the Lewis game with different forms of forgetting. (a) presents the $\rho$ across all phases of iterated learning; (b) zooms into the first 40000 steps to illustrate that $\rho$ improves in two stages for every generation: first during reinitialization and imitation (forget), but it is generally during interaction that the sender starts outperforming the previous generation's $\rho$ (relearn). (c) plots $\rho$ at the end of each generation for the ease-of-teaching setting.

improved performance in iterated learning is primarily driven by the interaction (retraining) phase. The retraining phase reinforces features that are consistently learned, and we conjecture that these correspond to the compositional mappings. We show in Figure 4b that the interaction phase is indeed primarily responsible for the improvement in $\rho$. By looking at $\rho$ during both the imitation and interaction phases using optimal hyperparameters under the setting in Ren et al. (2020), we see that increases in $\rho$ occur during the interaction phase, while the optimal imitation phase does not even need to result in a higher $\rho$ than the end of the previous generation.

To show that the compositional components of the language are repeatedly learned and strengthened during retraining, we visualize the learned mappings from each input dimension to message dimension in Appendix A4. We see that the same compositional mappings are retained across generations while the non-compositional aspects fluctuate and decrease in magnitude. We also observe that the probability of compositional mappings increase with increasing number of generations, illustrating the strengthening effects of iterative retraining.

## 6 CONCLUSION

We introduce *forget-and-relearn* as a general framework to unify a number of seemingly disparate iterative algorithms in the literature. Our work reveals several insights into what makes iterative training a successful paradigm. We show that various forms of weight perturbation, commonly found in iterative algorithms, disproportionately forget undesirable information. We hypothesize that this is a key factor to the success of these algorithms, and support this conjecture by showing that how well we can selectively forget undesirable information corresponds to the performance of the resulting algorithms. Although we focus discussion on the forgetting of undesirable information, our analysis in Section 5 implies that this is in service of preserving desirable information. We demonstrate that the relearning stage amplifies the unforgotten information, and distills from it features that are consistently useful under different initial conditions induced by the forgetting step.

This work has implications in understanding and designing new iterative algorithms. It places emphasis on *what is forgotten* rather than what is learned, and demonstrates that designing new methods for targeted information removal can be a fruitful area of research. It can often be easier to define and suppress unwanted behavior than to delineate good behavior, and we illustrate two ways of doing so. Our results suggest that there exists a *Goldilocks zone of forgetting*. Forget too little and the network could easily retrain to the same basin; forget too much and the network would fail to accumulate progress over multiple relearning rounds. Where this zone lies raises interesting questions in our understanding of neural network training that can be explored in future work. Finally, the idea that features that are consistently useful under different learning conditions is reminiscent of the line of work in invariant predictions (Arjovsky et al., 2019; Mitrovic et al., 2021; Zhang et al., 2021; Parascandolo et al., 2021). Understanding this dynamic in the iterative regime may point the way to creating algorithms that capture the same benefit without the expensive iterative training process.

ACKNOWLEDGMENTS

The authors would like to thank Kevin Guo, Jason Yosinski, Sara Hooker, Evgenii Nikishin, and Ibrahim Alabdulmohsin for helpful discussions and feedback on this work. The authors would also like to acknowledge Ahmed Taha, Fushan Li, and Yi Ren for open sourcing their code, which this work heavily leverages. This work was financially supported by CIFAR, Hitachi, and Samsung.

REPRODUCIBILITY

We make our code available at `https://github.com/hlml/fortuitous_forgetting`.

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

# A  APPENDIX

## A1  PARTIAL WEIGHT PERTURBATION AS TARGETED FORGETTING

### A1.1  DETAILS ON WEIGHT PERTURBATION METHODS

We can categorize this class of forgetting method in terms of both a *mask criteria* and a *mask action*. The mask criteria is a function $m : \mathbb{R}^d \to \{0, 1\}^d$, which determines which weights will be changed during the forgetting step by producing a binary mask with the same dimensions as the parameters of the neural network. The mask action $s : \mathbb{R}^d \to \mathbb{R}^d$ determines how the replacement values of the weights are computed. The mask-1 action $s_1$ operates on weights with mask value 1, and mask-0 action $s_0$ operates on weights with mask value 0. For a neural network parameterized by $\mathbf{\Theta}$, forgetting through partial weight perturbation can then be defined by the update rule

$$\mathbf{\Theta} \leftarrow m(\mathbf{\Theta}) \odot s_1(\mathbf{\Theta}) + (1 - m(\mathbf{\Theta})) \odot s_0(\mathbf{\Theta}). \tag{2}$$

Table A1: Types of weight perturbations for image classification experiments. Random mask criteria means that masks are generated randomly, and IMP-style mask criteria equals to 1 for corresponding weights with large final magnitudes. Mask-1 Action refers to the operation on weights with corresponding mask value of 1, and Mask-0 Action refers to the operation on weights with corresponding mask value of 0.

| Method | Mask Criteria | Mask-1 Action | Mask-0 Action |
|---|---|---|---|
| KE-style Random Reinit | Random | Identity | Reinitialize |
| IMP-style Weight Rewind | $|\mathbf{\Theta}_{final}|$ | Rewind | Set to 0 |

Table A2: A example of a compositional language adopted from Li & Bowling (2019).

|  | black | blue | green | grey | pink | purple | red | yellow |
|---|---|---|---|---|---|---|---|---|
| circle | bc | lc | gc | rc | pc | uc | dc | yc |
| square | bs | ls | gs | rs | ps | us | ds | ys |
| star | ba | la | ga | ra | pa | ua | da | ya |
| triangle | bt | lt | gt | rt | pt | ut | dt | yt |

Table A3: A example of a non-compositional language adopted from Li & Bowling (2019).

|  | black | blue | green | grey | pink | purple | red | yellow |
|---|---|---|---|---|---|---|---|---|
| circle | ys | gc | lc | pa | bc | ut | ua | rs |
| square | ga | la | yc | ra | dc | pt | rc | dt |
| star | yt | ls | ba | bs | bt | lt | uc | gs |
| triangle | pc | ps | da | ds | rt | us | ya | gt |

### A1.2  EXTENSION TO CIFAR-10 AND IMAGENET

We extend our analyses of partial weight perturbation to CIFAR-10 (Krizhevsky et al., 2009) on a 4-layer convolutional model (Conv4) and a ResNet18 model, and to ImageNet (Deng et al., 2009) on ResNet50. All CIFAR-10 models are evaluated using a 50% reset rate. We find reliable trends for both KE-style and IMP-style forgetting for all datasets.

For CIFAR-10, we follow the exact same procedure as the MNIST experiments. Following the setup in Frankle & Carbin (2019), Conv4 model is trained using the Adam optimizer, and ResNet18

Table A4: Types of weight perturbations for compositionality experiments. Random mask criteria means that masks are generated randomly, and weight mask criteria equals to 1 for corresponding weights with large final magnitudes. Frequency indicates how often masks are resampled. Mask-1 Action refers to the operation on weights with corresponding mask value of 1, and Mask-0 Action refers to the operation on weights with corresponding mask value of 0.

| Method | Mask Criteria | Frequency | Mask-1 Action | Mask-0 Action |
|---|---|---|---|---|
| Random Reinit | Random | Per Iter | Identity | Reinitialize |
| Random Zero | Random | Per Iter | Identity | Set to 0 |
| Same Weight Reinit | $|\Theta_{init}|$ | Once | Identity | Reinitialize |
| Same Weight Zero | $|\Theta_{init}|$ | Once | Identity | Set to 0 |

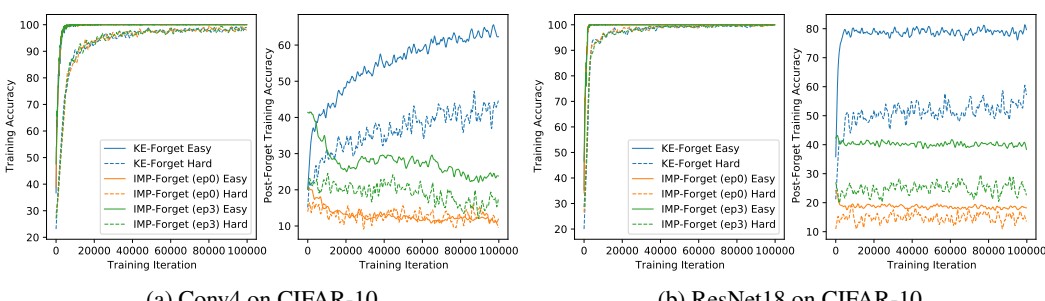

(a) Conv4 on CIFAR-10        (b) ResNet18 on CIFAR-10

Figure A1: The left panels in (a) and (b) show training accuracy of easy and hard example groups for different types of weight perturbation. The right panels in (a) and (b) show the accuracy of each example group for the same model with weight perturbation applied. Same colors represent the same forgetting operation, and dashed lines represent the accuracy on the hard examples. Results are averaged over 5 runs. Lowess smoothing is applied for visual clarity.

is trained using SGD. We train on 10% of training data for fast convergence. Results on easy and hard examples are shown in Figure A1. We find that KE-style forgetting results in consistent and significant difference between easy and hard examples. However, the IMP-style forgetting operation renders the post-forgetting accuracy close to chance for CIFAR-10 models, and there is only a small difference between easy and hard example groups. In the lottery ticket literature, it has long been observed that "late resetting" is needed to get performant sparse models as datasets and models get larger (Frankle et al., 2019). In late resetting, we would rewind kept weights to the value a few epochs after training, rather than to the initial untrained values. We perform a variant of IMP-style forgetting where weights are rewound to epoch 3. We find that IMP-style forgetting with late resetting significantly increases accuracy for both groups, and much more so for the easy group than the hard group. We also show the same trend with true and random labeled example groups in Figure A2. These results demonstrate that IMP with late resetting achieves more targeted forgetting of undesirable information, which can be seen through the larger gaps between example groups. These observations suggest another possible reason for the increased performance when using late resetting: IMP with late resetting better leverages the benefits of forget-and-relearn.

For ImageNet experiments, given computational constraints, we use a pretrained ResNet50 model on ImageNet and separate examples into easy and hard groups based on output margin. We only use training examples that are correctly classified as to not disadvantage the difficult examples. This means that both group has 100% accuracy pre-forgetting. We then perform various forgetting operations on top of the pretrained model and measure the accuracy of the two groups post-forgetting. For KE-style forgetting, we randomly reinitialize 20% of weights on the same pretrained model and aggregate results over 5 samples. For IMP-style forgetting, we set varying percentages of small weights to zero as a proxy given we are using a pretrained model and do not have access to initial or intermediate weights. Results are shown in Table A5, and we find that hard examples are more severely affected in all cases.

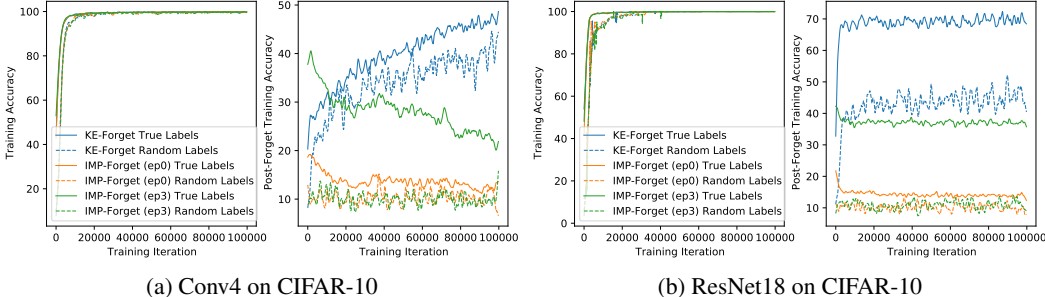

(a) Conv4 on CIFAR-10                          (b) ResNet18 on CIFAR-10

Figure A2: The left panels in (a) and (b) show training accuracy of true and mislabeled example groups for different types of weight perturbation. The right panels in (a) and (b) show the accuracy of each example group for the same model with weight perturbation applied. Same colors represent the same forgetting operation, and dashed lines represent the accuracy on the mislabeled examples. Results are averaged over 5 runs. Lowess smoothing is applied for visual clarity.

Table A5: Weight perturbations on ResNet50 for ImageNet. Using a pretrained model, we filter out the ImageNet training examples that are wrongly classified by the pretrained model. Hard examples are the 10% of remaining training examples with the largest output margin, and easy examples are the remaining training examples. We measure the accuracy of the two groups after performing various forgetting operations on the pretrained models. Random reinit values are the mean and standard error over 5 samples.

| Method | Easy Example Accuracy | Hard Example Accuracy |
|---|---|---|
| Random Reinit 20% of Weights | $45.6\% \pm 0.7\%$ | $17.9\% \pm 0.3\%$ |
| Set Small Weights to Zero (50%) | 89.8% | 47.3% |
| Set Small Weights to Zero (40%) | 97.9% | 65.5% |
| Set Small Weights to Zero (30%) | 99.7% | 80.1% |
| Set Small Weights to Zero (20%) | 100% | 90.3% |

## A2   Targeted Forgetting Improves Performance

### A2.1   Architectures and Training Details

All networks are trained with stochastic gradient descent (SGD) with momentum of $0.9$ and weight decay of $10^{-4}$. We also use a cosine learning rate schedule (Loshchilov & Hutter, 2017). Taha et al. (2021) use an initial learning rate of $0.256$, but we find that a smaller learning rate than what is used in Taha et al. (2021) to be beneficial for certain datasets, thus we consider a learning rate in $\{0.1, 0.256\}$ for all experiments and report the setting with the better validation performance. Models are trained with a batch size of $32$ for $200$ epochs each generation. No early stopping is used.

For the layer threshold $L$, we determine the value through hyperparameter tuning on a validation set. In the settings we consider, we find that it is sufficient to sweep through the starting layer or middle layer of model blocks, beginning from the second block. This allows us to search for $L$ within a restricted set of 6 of fewer values, making it practical even in deep models. All ResNet18 models use a default layer threshold of $10$ (start of block 3), with exception of the Flower dataset which uses a threshold of $14$ (start of block 4) on the Smth experiments. All DenseNet169 models use a default layer threshold of $40$ (start of denseblock 3), except for the Aircraft dataset which uses a threshold of $68$ (middle of denseblock 3).

Table A6: Summary of the five datasets used in Section 4.1, adopted from Taha et al. (2021).

|  | Classes | Train Size | Val Size | Test Size | Total Size |
|---|---|---|---|---|---|
| Flower (Nilsback & Zisserman, 2008) | 102 | 1020 | 1020 | 6149 | 8189 |
| CUB (Wah et al., 2011) | 200 | 5994 | N/A | 5794 | 11788 |
| Aircraft (Maji et al., 2013) | 100 | 3334 | 3333 | 3333 | 10000 |
| MIT67 (Quattoni & Torralba, 2009) | 67 | 5360 | N/A | 1340 | 6700 |
| Stanford-Dogs (Khosla et al., 2011) | 120 | 12000 | N/A | 8580 | 20580 |

Table A7: Later Layer Forgetting on Tiny-ImageNet. Results are test accuracy averaged over 3 runs and reported for hyperparameter settings with best validation performance. N3 and N10 indicate the additional number of training generations on top of the baseline model. The long baseline scales the step LR schedule based on the additional epochs.

| Method | Tiny-ImageNet |
|---|---|
| Smth (N1) | 54.37 |
| Smth long (N3) | 51.16 |
| Smth + LLF (N3) **(Ours)** | **56.12** |
| Smth long (N10) | 49.27 |
| Smth + LLF (N10) **(Ours)** | **56.92** |

## A2.2  DATASET SUMMARY

Taha et al. (2021) uses different resizing of images for different datasets, and the specific resizing parameters are not specified in the paper. For consistency, we resize all datasets to $(256, 256)$. We use a validation split to tune hyperparameters and report results on the test set when available.

## A2.3  EXTENSION TO LARGER DATASETS AND MODELS

In this section, we evaluate LLF on larger datasets and models than what was used in the core comparison with KE. We train ResNet50 on Tiny-ImageNet (Le & Yang, 2015), which consists of 200 classes with 500 training examples each. We adopt the training setup from Ma et al. (2021) and reset layers starting from the third block ($L = 23$) during LLF. As illustrated in Table A7, we find LLF to outperform the baselines on Tiny-ImageNet.

Furthermore, we also adopt two highly competitive baselines for CIFAR-10 and CIFAR-100 (Krizhevsky et al., 2009) in the literature: WideResNet-28-10 (Zagoruyko & Komodakis, 2016) and DenseNet-BC ($k = 12$) (Huang et al., 2017). CIFAR-10 contains 10 classes with 5000 training examples each, while CIFAR-100 contains 100 classes with 500 examples each. For WideResNet-28-10, we reset from the second block ($L = 10$), and for DenseNet-BC, we reset only the fully-connected output layer ($L = 99$) during LLF. We report our results in Table A8. In these settings, we do not see any improvements from LLF over our baselines.

For these dataset extensions, we follow the same simple heuristic that was used for the five small datasets in the main text. Namely, we take existing training setups and hyperparameters from the literature as is and treat it as one generation, and search for $L$ over a restricted set of values (6 or fewer). Better results may be obtained through finer hyperparameters tuning.

## A2.4  COMPARISON TO LW

Concurrent work (Alabdulmohsin et al., 2021) studies various types of iterative reinitialization approaches and finds that reinitializing entire layers produced the best performance. They propose a layer-wise reinitialization scheme, proceeding from bottom to top and reinitializing one fewer layer each generation. In addition, Alabdulmohsin et al. (2021) performs weight rescaling and activation normalization for each layer that is not reinitialized. We refer to this method as LW and add

Table A8: Later Layer Forgetting on CIFAR-10 and CIFAR-100 using WideResNet-28-10 (WRN) and DenseNet-BC (DN). Results are test accuracy averaged over 3 runs and reported for hyperparameter settings with best validation performance. We consider three additional training generations on top of the baseline model. The long baseline scales the step LR schedule based on the additional epochs.

| Method | CIFAR-10 | CIFAR-100 |
|---|---|---|
| WRN Smth (N1) | 96.09 | 81.20 |
| WRN Smth long (N3) | **96.32** | **81.29** |
| WRN Smth + LLF (N3) (**Ours**) | 95.91 | 80.95 |
| DN Smth (N1) | 95.21 | 77.12 |
| DN Smth long (N3) | **95.68** | 78.12 |
| DN Smth + LLF (N3) (**Ours**) | 95.52 | **78.13** |

a comparison to their method in Table 1. We find that LLF outperforms LW across all datasets we consider. However, we note that LLF introduces an additional hyperparameter for the layer threshold, which LW does not require. Additionally, we find that under our experimental settings, several choices proposed in LW actually harms performance. We remove the addition of weight rescaling and normalization, and do not reinitialize batchnorm parameters, in order to achieve the best performance under the LW schedule. The forget-and-relearn hypothesis suggests that it is unnecessary to reinitialize the first few layers. Doing so in LW hinders their progress in the first few generations of training, and also leads to poorer generalization performance even after completing the full sequential schedule.

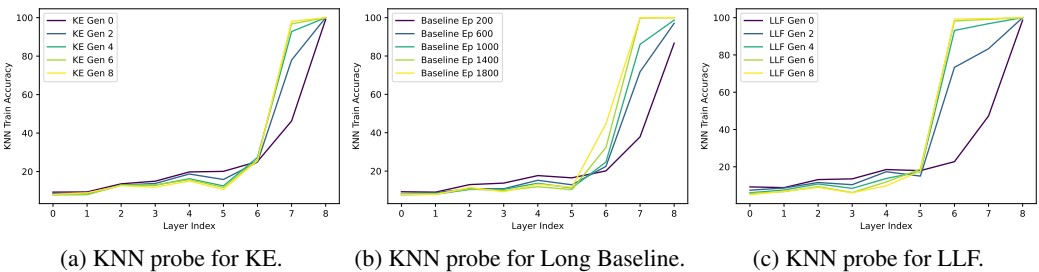

(a) KNN probe for KE.  (b) KNN probe for Long Baseline.  (c) KNN probe for LLF.

Figure A3: KNN probe train accuracy for various ResNet18 models trained on Flower. The layers shown are the end of each block and after the softmax operation. We use K=3 to compute accuracy using the activations from each layer. The different curves in each panel show that overall prediction depth (as approximated by average KNN accuracy across layers) decreases with more training. LLF in (c) shows the most significant change, with the KNN probe performing near perfect at layer 6, which is earlier than the other two methods.

## A3  DETAILS ON PARTIAL BALANCED FORGETTING

Our experimental settings follows Li & Bowling (2019). We use one-layer LSTMs with a hidden size of 100 to parameterize both the sender and receiver agents. We train using batch size of 100, and use the Adam optimizer (Kingma & Ba, 2014) with a learning rate of 0.001. The language contains a vocabulary size of 8 and a message length of 2. We use 4 SHAPE and 8 COLOR attributes, and the object set consist of all 32 combinations of these attributes.

The agents are trained with REINFORCE and entropy regularization using the following objectives:

$$\nabla_{\theta^S} J = \mathbb{E}_{\pi^S, \pi^R}[R(\hat{t}, t) \cdot \nabla_{\theta^S} \log \pi^S(m \mid t)] + \lambda^S \cdot \nabla_{\theta^S} H[\pi^S(m \mid t)]$$

Table A9: Comparing KE and Later Layer Forgetting for ResNet18. Results are test accuracy averaged over 3 runs and reported for hyperparameter settings with best validation performance. KE experiments use KELS split with a split rate of $0.8$. LLF uses $L \in \{10, 14\}$, corresponding to block 3 and 4 in ResNet18. N3, N8, N10 indicate the additional number of training generations on top of the baseline model. LLF consistently outperforms all other methods.

| Method | Flower | CUB | Aircraft | MIT | Dog |
|---|---|---|---|---|---|
| Smth (N1) | 51.02 | 58.92 | 57.16 | 56.04 | 63.64 |
| Smth long (N3) | 59.51 | 66.03 | 62.55 | 59.53 | 65.39 |
| Smth + KE (N3) | 57.95 | 63.49 | 60.56 | 58.78 | 64.23 |
| Smth + LLF (N3) **(Ours)** | **63.52** | **70.76** | **68.88** | **63.28** | **67.54** |
| Smth long (N10) | 66.89 | 70.50 | 65.29 | 61.29 | 66.19 |
| Smth + KE (N10) | 63.25 | 66.51 | 63.32 | 59.58 | 63.86 |
| Smth + LLF (N10) **(Ours)** | **70.87** | **72.47** | **70.82** | **64.40** | **68.51** |
| Smth + LW (N8) | 62.84 | 70.58 | 67.36 | 61.24 | 66.58 |
| Smth + LW NoNorm NoRescale (N8) | 65.97 | 71.15 | 67.97 | 63.13 | 66.96 |
| Smth + LW(-BN) (N8) | 63.67 | 70.10 | 66.79 | 60.10 | - |
| Smth + LW(-BN) NoNorm NoRescale (N8) | 68.43 | 70.87 | 69.10 | 61.67 | 66.97 |
| Smth + LLF (N8) **(Ours)** | **69.48** | **72.30** | **70.37** | **63.58** | **68.45** |
| CS-KD (N1) | 57.57 | 66.61 | 65.18 | 58.61 | 66.48 |
| CS-KD long (N3) | 64.44 | 69.50 | 65.23 | 57.16 | 66.43 |
| CS-KD + KE (N3) | 63.48 | 68.76 | 67.16 | 58.88 | 67.05 |
| CS-KD + LLF (N3) **(Ours)** | **67.20** | **72.58** | **71.65** | **62.41** | **68.77** |
| CS-KD long (N10) | 68.68 | 69.59 | 64.58 | 56.12 | 64.96 |
| CS-KD + KE (N10) | 67.29 | 69.54 | 68.70 | 57.61 | 67.11 |
| CS-KD + LLF (N10) **(Ours)** | **74.68** | **73.51** | **72.01** | **62.89** | **69.20** |
| CS-KD + LW (N8) | 65.80 | 70.94 | 66.37 | 57.96 | 66.74 |
| CS-KD + LW NoNorm NoRescale (N8) | 69.96 | 70.91 | 69.53 | 60.22 | 67.47 |
| CS-KD + LW(-BN) (N8) | 69.54 | - | 68.87 | - | - |
| CS-KD + LW(-BN) NoNorm NoRescale (N8) | **73.72** | 71.81 | 70.82 | 59.18 | 68.09 |
| CS-KD + LLF (N8) **(Ours)** | 73.48 | **73.47** | **71.95** | **62.26** | **69.24** |

$$\nabla_{\theta^R} J = \mathbb{E}_{\pi^S, \pi^R}[R(\hat{t}, t) \cdot \nabla_{\theta^R} \log \pi^R(\hat{t} \mid m, c)] + \lambda^R \cdot \nabla_{\theta^R} H[\pi^R(\hat{t} \mid m, c)]$$

We use $\lambda^S = 0.1$ and $\lambda^R = 0.1$ in all our experiments, which differs slightly from Li & Bowling (2019) that use $\lambda^R = 0.05$, however we observe that the difference is small between these settings. We train the "no reset" and "reset receiver" baselines for 6000 iterations per generation for 50 generations, following the hyperparameters in Li & Bowling (2019). We train our PBF method with "same weight reinit" for 3000 iterations per generation for 100 generations for best performance. However we note that even using the same 50 generation hyperparameter, PBF significantly outperforms its baselines.

## A4 EMERGENT LANGUAGE VISUALIZATIONS

In this section, we aim to visualize the languages learned by the sender in a Lewis game in the settings of Ren et al. (2020) and Li & Bowling (2019). Despite all the runs having near-perfect training accuracy , the degree of compositionality exhibited by the emergent languages varies significantly depending on the use of forget-and-retrain as evident in Figure 4. In both these settings, each object has two attributes that can take values in $\mathcal{A}_1 = \{1, \ldots, r_1\}$ and $\mathcal{A}_2 = \{1, \ldots, r_2\}$ respectively, and each message has two tokens that can take values in $\mathcal{M}_1 = \{1, \ldots, s_1\}$ and $\mathcal{M}_2 = \{1, \ldots, s_2\}$ respectively. Let us represent $a_i^{(j)}$ as 1 if attribute $i$ has the value $j$, and 0 otherwise. Similarly, we represent $m_i^{(j)}$ as 1 if message token $i$ has the value $j$, and 0 otherwise. When $r_1 + r_2 = s_1 + s_2$

Table A10: Comparing KE and Later Layer Forgetting for DenseNet169. Results are test accuracy averaged over 3 runs and reported for hyperparameter settings with best validation performance. KE experiments use WELS split with a split rate of 0.7. LLF uses $L \in \{40, 68\}$. N3, N8, N10 indicate the additional number of training generations on top of the baseline model. LLF consistently outperforms all other methods.

| Method | Flower | CUB | Aircraft | MIT | Dog |
|---|---|---|---|---|---|
| Smth (N1) | 45.87 | 61.59 | 58.06 | 57.21 | 66.46 |
| Smth long (N3) | 59.39 | 70.87 | 67.55 | 62.29 | 68.82 |
| Smth + KE (N3) | 58.35 | 68.85 | 65.63 | 60.35 | 68.64 |
| Smth + LLF (N3) (**Ours**) | **62.31** | **71.97** | **70.53** | **64.60** | **70.19** |
| Smth long (N10) | 67.47 | 71.98 | 70.69 | 60.72 | 67.48 |
| Smth + KE (N10) | 65.19 | 70.20 | 67.47 | 60.77 | 68.62 |
| Smth + LLF (N10) (**Ours**) | **70.09** | **73.12** | **74.31** | **61.69** | **69.93** |
| CS-KD (N1) | 52.95 | 64.28 | 64.87 | 57.61 | 66.90 |
| CS-KD long (N3) | 60.03 | 63.80 | 67.68 | 57.21 | 67.01 |
| CS-KD + KE (N3) | 63.33 | 66.78 | 68.47 | 58.81 | 68.49 |
| CS-KD + LLF (N3) (**Ours**) | **65.43** | **73.54** | **72.22** | **61.74** | **70.61** |
| CS-KD long (N10) | 63.84 | 63.58 | 67.22 | 55.49 | 64.69 |
| CS-KD + KE (N10) | 69.24 | 66.34 | 69.51 | 58.33 | 67.73 |
| CS-KD + LLF (N10) (**Ours**) | **72.23** | **74.20** | **73.30** | **63.18** | **71.07** |

(*i.e.* the total number of input attribute values is equal to the total number of message token values), a compositional language with maximum topographic similarity ($\rho$) would have each input attribute activation $a_k^{(l)}$ mapped to a unique message token activation $m_p^{(q)}$.

Thus, we can represent the inputs and the messages both as concatenations of two one-hot vectors $\boldsymbol{a} = \{a_1^{(1)}, \ldots, a_1^{(r_1)}, a_2^{(1)}, \ldots, a_2^{(r_2)}\}$ and $\boldsymbol{m} = \{m_1^{(1)}, \ldots, m_1^{(s_1)}, m_2^{(1)}, \ldots, m_2^{(s_2)}\}$ respectively. In Figures A4-A8, we illustrate the mappings from input dimensions $\boldsymbol{a}$ to message dimensions $\boldsymbol{m}$. Visualized values closer to 1 indicate a more consistent mapping across the entire dataset, indicating a compositional mapping. Concretely, for each active input dimension $a_k^{(l)}$, we vary $\mathcal{A}_{\neq k}$ across all its values and collect the corresponding message token probabilities $\hat{\boldsymbol{m}} = \{\hat{m_1}^{(1)}, \ldots, \hat{m_1}^{(s_1)}, \hat{m_2}^{(1)}, \ldots, \hat{m_2}^{(s_2)}\}$ produced by the sender via two softmax operations. Then, the mean of these message probability vectors $\mathbb{E}_{\mathcal{A}_{\neq k}}[\hat{\boldsymbol{m}}]$ across $\mathcal{A}_{\neq k}$ is used as the row corresponding to input dimension $a_k^{(l)}$. For a perfectly compositional mapping, the visualized matrix resembles a permutation matrix.

For iterated learning in Figure A4 and PBF in Figure A6, we see the emergence of a compositional mapping. The compositional component is retained across generations whereas the non-compositional components are disproportionately affected in the forgetting step and are more likely to get remapped. Finally, during the retraining phases, we see the compositional mappings get strengthened.

## A5 CONNECTIONS TO OTHER TYPES OF FORGETTING

In the continual learning literature, catastrophic forgetting (McCloskey & Cohen, 1989; Ratcliff, 1990) refers to the decrease in performance on older tasks due to the learning of new tasks. There is a large body of literature focused on avoiding catastrophic forgetting (Ratcliff, 1990; Robins, 1995; French, 1999; Kirkpatrick et al., 2017; Serra et al., 2018). In this work, we view forgetting in a positive light, to say that it aides learning. It is conceivable that catastrophic forgetting can even be leveraged to help learning with a well-chosen secondary task in a forget-and-relearn process. It is worth noting that the continual learning problem is in some sense the opposite of what we consider

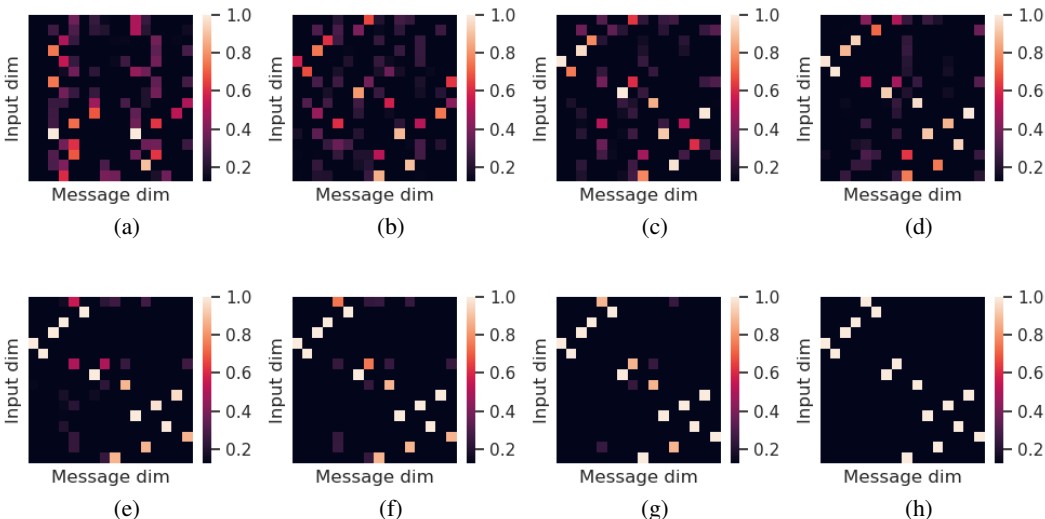

Figure A4: Mappings from input dimension to message dimension learned by the sender using **iterated learning** in the setting of Ren et al. (2020), logged after every generation for the first 32200 training iterations across all phases. A compositional language appears as a permutation matrix, where each input dimension maps to a unique message dimension.

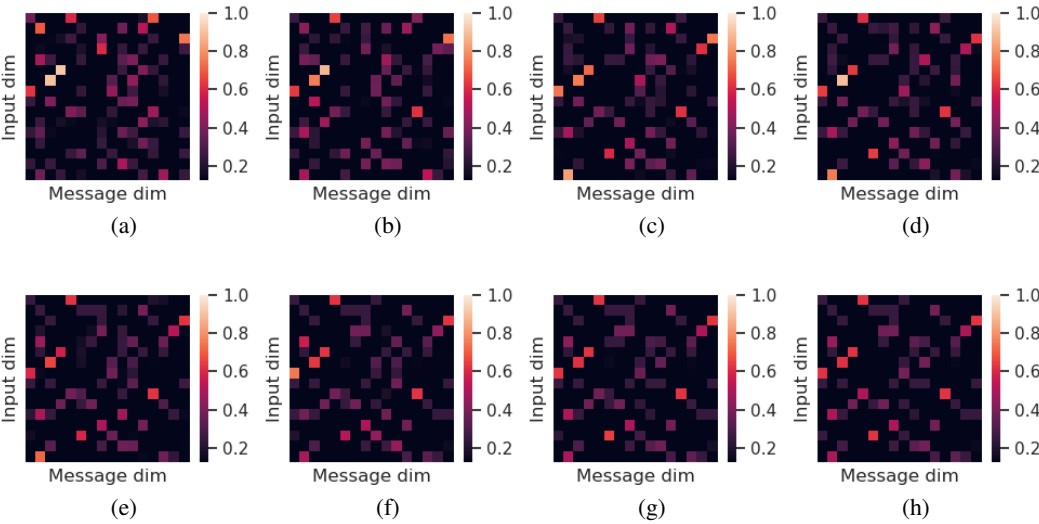

Figure A5: Mappings from input dimension to message dimension learned by the sender **without any form of forget-and-retrain** in the setting of Ren et al. (2020), logged uniformly for the first 33000 training iterations. A compositional language would have appeared as a permutation matrix, where each input dimension maps to a unique message dimension.

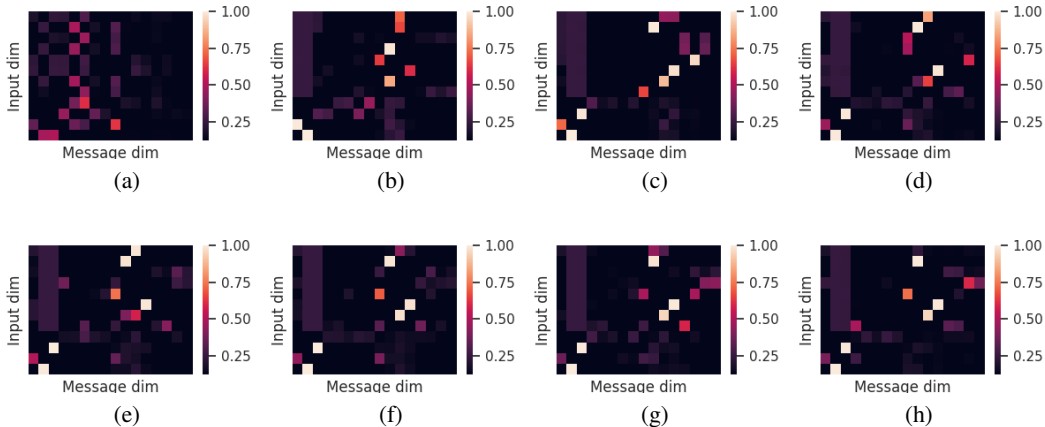

Figure A6: Mappings from input dimension to message dimension learned by the sender using **Partial Balanced Forgetting (PBF)** in the setting of Li & Bowling (2019), logged after every two generations for the first 48000 training iterations. A more compositional language has unique non-overlapping message dimension activations for different input dimensions.

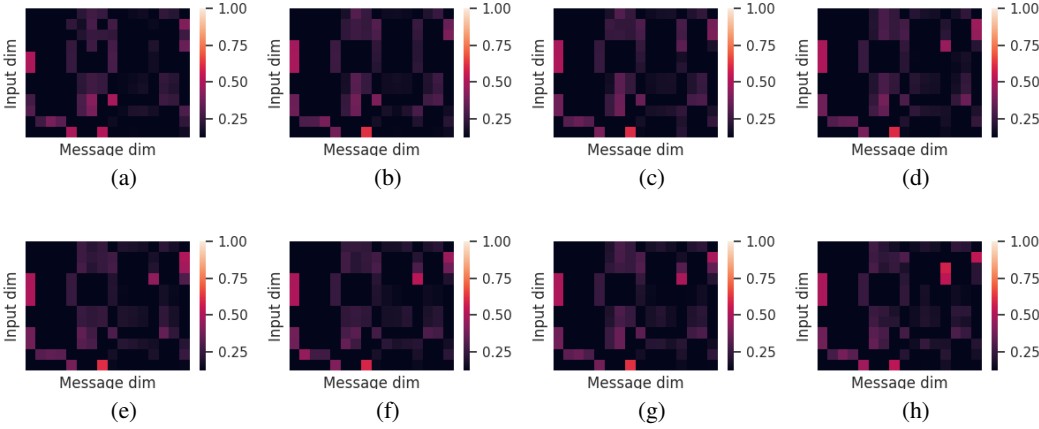

Figure A7: Mappings from input dimension to message dimension learned by the sender by **resetting the receiver periodically** following the setting of Li & Bowling (2019), logged after every generation for the first 48000 training iterations. A more compositional language has unique non-overlapping message dimension activations for different input dimensions.

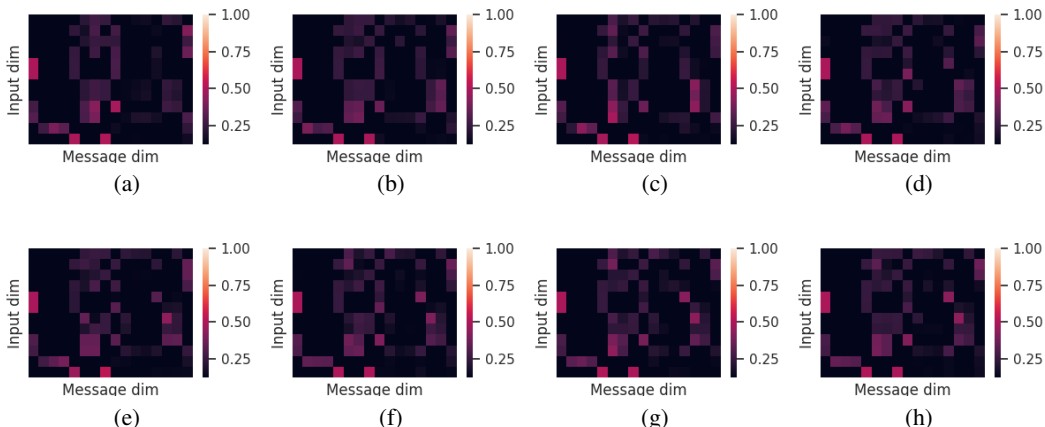

Figure A8: Mappings from input dimension to message dimension learned by the sender **without any form of forget-and-retrain** in the setting of Li & Bowling (2019), logged uniformly for the first 48000 training iterations. A more compositional language has unique non-overlapping message dimension activations for different input dimensions.

in forget-and-relearn. Continual learning seeks to learn different tasks without forgetting old ones, while we explicitly assume that the tasks are available for iterative relearning.

Shwartz-Ziv & Tishby (2017) studies the progression of mutual information between each layer and the input and target variables throughout the training of deep neural networks. They reveal that the converged values lie close to the information bottleneck theoretical bound, which implies a decrease in the mutual information with the input. In a sense, the information bottleneck leads to the "forgetting" of unnecessary information. It would be interesting to explore whether there are fundamental similarities between the information loss caused by SGD training dynamics and the effect resulting from forget-and-relearn.

Barrett & Zollman (2009) studies forgetting in the Lewis game from the perspective of suboptimal equilibria. They show that using a learning rule that remembers the entire past (i.e. Herrnstein reinforcement learning), the model converges to a worse equilibrium than learning rules that discard past experience. Yalnizyan-Carson & Richards (2021) also studies the benefits of forgetting in the reinforcement learning formulation. These works further point to the possibility of designing forgetting mechanisms that are built in to the learning process itself.

Although we claim many iterative training algorithms as instances of forget-and-relearn, there are also many that we do not, even though they may be related. Algorithms like born-again networks (Furlanello et al., 2018) and self-training with noisy students (Xie et al., 2020) rely on iterative self-distillation, and do not have distinct forgetting and relearning stages. Instead, knowledge is passed to the next generation through pseudo-labels. A number of papers have tried to understand why students trained through self-distillation often generalize better than their teachers (Phuong & Lampert, 2019; Tang et al., 2020; Allen-Zhu & Li, 2020; Stanton et al., 2021); perhaps viewing these methods through the lens of forgetting undesirable information offers an alternative path.

