# OpenReview forum: "Fortuitous Forgetting in Connectionist Networks"
_ICLR.cc/2022/Conference — ICLR 2022 Poster_

### Official Review · Reviewer_hRdL · 2021-11-02

**Correctness:** 3
**Technical Novelty And Significance:** 3
**Empirical Novelty And Significance:** 3
**Recommendation:** 6
**Confidence:** 3

**Main Review:**


The work presents an alternative hypothesis to the idea of forgetting behavior in models. Indeed, it is an interesting proposal. Forgetting to leave room to better capture existing features and relearning the more difficult samples is a curious perspective.

Strengths:

* The ideas in this work are novel to the best of my knowledge.
* The work is clear, well written and enjoyable to read.
* The experiments nicely demonstrate the paradigm

Weaknesses:

* The datasets in the paper are not referenced. Please, include references, and include in the text a description of them, like sizes, etc.
* The work is shown for two tasks in their limited setups. This poses a question of whether the hypothesis can be proven and is general. The work does not prove the hypothesis, which jeopardizes the generality of the method. Unfortunately, a formal proof is not included.
* Still, if we assume the hypothesis to be generally true, how can the reader understand to apply the paradigm to any other problem?
* A few experimental details are not discussed. For example, how is L chosen to be within the specific values in the big networks? A better way would be to show the performance as function of L.

Questions to the authors:
- The improvements to iterative algorithms obtained under “forget-to-relearn” were handled based on properties from the model and its learning. How can we know what features, layers, etc. should be reset in the general case? If we don’t know a-priori where or how the tail of features is learned, how can we improve training further? I would appreciate a general discussion from the authors added to the paper on this matter.
- In language, it has been proposed that a network could be pre-trained for shorter sequences and later on with longer ones (e.g., [1]). Could you discuss how the forget-relearn paradigm relates to such cases where the network is pre-trained and then extended to train on additional weights or inputs?
- Recent research in generalization of neural networks, suggests that overparameterization, first overfits and then a second dip allows for better generalization of models [2]. Are these two processes complementary or contradictory? Why?

[1] Devlin et al., BERT: Pre-training of Deep Bidirectional Transformers for Language Understanding, 2018.
[2] Belkin et al., Reconciling modern machine-learning practice and the classical bias–variance trade-off, 2019.


**Summary Of The Paper:**

The paper explores the forgetting effect as a positive effect for learning. The authors suggest that a “forgetting to relearn” hypothesis is the underlying mechanism. This mechanism removes unvaluable or weak features at forget time, while models relearn and reinforce the remaining features. Then the hypothesis is used to explain effects on iterative training algorithms and further suggest some improvements. The hypothesis is empirically evaluated by improving iterative algorithms in two cases: image classification, and language emergence via the Lewis game. In image classification, the authors propose later-layer forgetting (LLF) based on the idea that easier samples are decided in the first layers of a convolutional neural network, while difficult examples are learned in deeper layers. LLF reinitializes only the weights in deeper layers. The results over 5 datasets (Flower, CUB, Aircraft, MIT, Dog) show improvements over long training and other restarting methods, as well as with ResNet and DenseNet as models.
In language, the manuscript tests the Lewis game with a mixture of compositional and non-compositional language as the easy and difficult samples, respectively. Also, in this case, the authors come up with Partial Balanced Forgetting (PBF) which partially reinitializes both the sender and the receiver in the game, showing better results than previous hypothesized ideas about this game. Last, they test for the learning aspects during retraining under the forget and relearn paradigm, such as the strengthen of the model during retraining.


**Summary Of The Review:**

Despite the main idea being counterintuitive at first glance, the topic becomes clearer and more interesting with the reading and the experiments, nicely explaining previous works. Therefore, I believe that the ideas have merit, however there is no theoretical proof that proves the hypothesis. There are some impracticalities of this work, that I would appreciate if the authors can address.

---

> ### Author Response · Authors · 2021-11-18
> **Response to review [2/2]**
>
>
>
>
>
> > Still, if we assume the hypothesis to be generally true, how can the reader understand to apply the paradigm to any other problem?
> > The improvements to iterative algorithms obtained under “forget-to-relearn” were handled based on properties from the model and its learning. How can we know what features, layers, etc. should be reset in the general case? If we don’t know a-priori where or how the tail of features is learned, how can we improve training further? I would appreciate a general discussion from the authors added to the paper on this matter.
>
> Thank you for raising these questions, we think that they point to key follow-ups in this line of work. The forget-and-relearn perspective offers an actionable hypothesis for designing iterative algorithms. It suggests that we should aim to remove undesirable information from the model through well-designed forgetting operations. However, the best way of doing so will always depend on the particular task at hand. As we saw in the two settings studied in the paper, what’s undesirable is different for different tasks. For generalization on small datasets, we used difficult examples as a proxy for undesirable information. For language emergence, we viewed non-compositional languages as undesirable. We can also imagine use cases where there are biases that we want to eliminate from the model, which would again lead to problem-specific definitions of undesirability. Moreover, forgetting need not be done using localized weight reinitializations. Our study of iterated learning provides a counterexample, where the forgetting operation uses a generational transmission of information through a learning bottleneck. This process removes information that takes longer to learn, which has been shown to be effective in many tasks requiring compositionality.
>
> For the problem of improving generalization in image classification, many of our results are inspired by prior works showing general phenomena in neural networks. The observation that later layers are more responsible for memorization [1], that models which are robust to weight perturbations show better generalization [2][3], and that pruning disproportionately affect the tails of the data distribution [4], are all general observations of neural networks. Thus, we can adapt similar methods to new datasets through hyperparameter tuning. Nonetheless, we caution practitioners that forgetting and relearning requires a delicate balance that depends on factors such as network capacity, task complexity,  and how smoothly the remaining information can adapt to new learning conditions induced by forgetting. Methods like LLF that work well in the small dataset setting may not be the most effective form of forgetting for large datasets.
>
> Finally, we speculate that a promising direction to tackle the problem more generally is to treat forgetting as a learned problem. This shifts the onus from identifying specific locations within the network responsible for undesirable features, to using training objectives designed to tease out undesirable information.
>
> > In language, it has been proposed that a network could be pre-trained for shorter sequences and later on with longer ones (e.g., [1]). Could you discuss how the forget-relearn paradigm relates to such cases where the network is pre-trained and then extended to train on additional weights or inputs?
>
> RIFLE [5] is an example of forget-and-relearn applied to the transfer learning setting, thus the idea is very much applicable to pretrained models. The point on BERT, if we understood you correctly, is related to forms of curriculum learning in general. Indeed, there could be interesting connections. We can speculate that the disproportionate forgetting of difficult examples may be doing some implicit forms of curriculum learning. Algorithms like LLF may be amplifying features associated with easy examples and delaying the learning of more difficult examples.
>
> >Recent research in generalization of neural networks, suggests that overparameterization, first overfits and then a second dip allows for better generalization of models [2]. Are these two processes complementary or contradictory? Why?
>
> To the best of our understanding so far, double descent and forget-and-relearn seem to be unrelated phenomena. We’d be very interested to hear any connections you may see between these ideas.
>
> Thank you again for raising these constructive questions and for appreciating our work. We hope that we were able to address your concerns, and we would welcome any further discussion if not.
>
> [1] https://arxiv.org/abs/2106.09647
>
> [2] https://arxiv.org/abs/1802.05296
>
> [3] https://arxiv.org/abs/2010.01412
>
> [4] https://arxiv.org/abs/1911.05248
>
> [5] https://arxiv.org/abs/2007.03349

---

> ### Author Response · Authors · 2021-11-18
> **Response to review [1/2]**
>
> Thank you for your time and effort in reviewing our paper. We are delighted that you found the ideas in this paper novel and interesting, and that our experimental results nicely support the paradigm. We would like to address the weaknesses and questions you’ve raised, and we’d be very open to further discussion.
>
> > The datasets in the paper are not referenced. Please, include references, and include in the text a description of them, like sizes, etc.
>
> Due to space constraints, this information is included in the appendix. Please see section A2.2 in the appendix.
>
> >A few experimental details are not discussed. For example, how is L chosen to be within the specific values in the big networks? A better way would be to show the performance as function of L.
>
> L is chosen through hyperparameter search on a validation set. Hyperparameter choices are discussed in the appendix (A2.1). We found that it is sufficient to search for L block-wise rather than layer-wise. DenseNet169, Resnet18, Resnet50 all compose of 4 blocks (with varying number of conv layers each block) and a FC layer. We found decent hyperparameters by only looking at the starting layer and the center layer of blocks 2,3,4, and FC, and the best L always fell between blocks 3 and 4 in our experiments. Thus, even in deep networks with 169 layers, the search for L can be very practical.
>
> As an example, we show the performance of DenseNet169 on Aircrafts at 10 generations, where layer 1 is the start and layer 15 is the middle of the block.
>
> |Reset start layer | Test Accuracy|
> |-------------------|----------------|
> |Block 2, layer 15 |62.7|
> |Block 3, layer 1 |66.5|
> |**Block 3, layer 15**| **71.2**|
> |Block 4, layer 1| 65.0|
> |Long baseline| 65.3|
> |KE |63.3|
>
> > The work is shown for two tasks in their limited setups. This poses a question of whether the hypothesis can be proven and is general. The work does not prove the hypothesis, which jeopardizes the generality of the method. Unfortunately, a formal proof is not included.
>
> This paper is indeed an empirical paper. We hope that theoretical advancements on this topic will be subjects of future study. Whether the hypothesis can be shown to apply in any setting is indeed an important and unanswered question. The two tasks we study, namely image classification and language emergence, are very different problems. Thus, although these demonstrations are limited in scope, we think they offer compelling evidence that forget-and-relearn isn’t an isolated phenomenon.

---

### Official Review · Reviewer_MVhg · 2021-11-02

**Correctness:** 4
**Technical Novelty And Significance:** 3
**Empirical Novelty And Significance:** 4
**Recommendation:** 10
**Confidence:** 4

**Main Review:**

The paper is beautifully written and lets the reader smoothly build an understanding of recent DL regularization methods over the proposed forget-and-relearn foundation.
The scheme of a clearly stated hypothesis at the beginning that undergoes falsification attacks throughout the paper is well-realized and should set a trend.
The paper is technically sound, from the definition of the forgetting operation to the setup of the various experiments and their presentation. The math discourse flows.
Authors show great confidence in recent literature. All the paper is disseminated with well-discussed references that are valuable to the reader also beyond this specific work.

A relevant contribution of this work is the Later-Layer-Forgetting NN training method that substantially outperforms baselines and competitors in image classification. The results of the experiments in the entirely different scenario of language emergence give further evidence to the generality of the LLF approach.
Nevertheless, I believe that the core contribution of this work remains the forget-and-relearn interpretation of existing (and future) DL methods, which shows a remarkable explanatory potential - demonstrated by the straightforward derivation of the LLF method itself - and could positively animate the ML and DL debate as other proposed interpretations did in the past, such as the lottery ticket hypothesis.

I hope authors persist in this line of research in the future. I would undoubtedly follow new works in this direction. In this regard, I believe authors can integrate their conclusion with a richer discussion of future directions. I think there are many:
- The accuracy gains persist with larger datasets such as Imagenet? How many forget and relearn cycles we should perform? Can we simply wait for convergence also for big datasets?
- Can the forget-and-relearn paradigm help in refining our understanding of the continual learning problem? How should we use LLF in a continual learning scenario?
- What is the impact of LLF in deep reinforcement learning? For instance, the iterative training procedure of AlphaGo Zero that imposes to learn under a new condition at each generation could especially benefit from forgetting-and-relearn methods such as LLF.
- How LLF should be applied to other learning problems and architectures (e.g., transformers)?
- Do we register further gains in accuracy by gradually forgetting fewer layers in LLF?

**Summary Of The Paper:**

The authors propose forgetting as a key concept to understand the success of recent DL regularization methods such as Iterative Magnitude pruning. The authors articulate their view in the *forget-and-relearn* hypothesis and extensively test it through the paper. They leverage this understanding to shape a new simple forget-and-relearn regularization scheme that effectively reduces overfitting in image classification and promotes compositionality in language emergence problems.

**Summary Of The Review:**

This paper offers a new hypothesis to explain the success of several recent DL methods. This hypothesis is well supported by experiments and offers a straightforward method to reduce overfitting, effective in image classification and language emergence problems. I believe it could positively animate the ML and DL debate as other proposed interpretations did in the past, such as the lottery ticket hypothesis. In my opinion, this article should be highlighted at the conference.

---

> ### Comment · Reviewer_SWHW · 2021-11-03
> **Is this really a 10 (strong accept / highlight)?**
>
> Since we disagree in our views on this paper, it would be insightful to discuss and compare our points of view.
>
> For instance, I agree with all the strengths you have listed, but I have a different view on how significant they are:
>
> > Later-Layer-Forgetting NN training method that substantially outperforms baselines and competitors in image classification
>
> For what I can tell, LLF outperforms one other forgetting-based technique on a number of *small* image classification tasks.
> To the best of my (limited) knowledge, the comparison is done in a not-very-competitive setting, as most of these datasets can be solved with significantly higher accuracy. For instance, based on [1], both CUB and Stanford Dogs can be solved with over 90% accuracy with transfer learning based methods.
>
> [Request to ACs: please make this comment visible to the reviewer when it is allowed. At the time of writing, I was unable to change its visibility scope.]
>
> To clarify, I do not doubt the validity of their experimental setup, but I would argue that solving image classification on extremely small datasets without transfer learning is a niche task with limited significance. Furthermore I'm afraid that not emphasizing this in the paper runs a risk of misleading some readers into thinking that the experimental setup is representative of a typical image classification task.
>
> > The scheme of a clearly stated hypothesis at the beginning that undergoes falsification attacks throughout the paper is well-realized and should set a trend.
>
> In my original review, I counted this "deviation from traditional structure" as a minor weakness. However, I see now that this scheme does have a merit.
>
> > The accuracy gains persist with larger datasets such as Imagenet? ; How LLF should be applied to other learning problems and  architectures (e.g., transformers)?
>
> Curiously, we both emphasize this in our reviews, but with opposite connotations. Your review commends the investigation that the paper already has and propose directions for future work. In contrast, I believe that the current experiments are too weak to support a **major** claim about how forgetting can improve DNNs in general -- and hence I believe that at least some of those additional experiments are necessary to fully support the claim.
>
> Please elaborate: do you believe that the existing evidence is sufficient comparable with what similar sudies[2,3] provide?
>
> [1] https://paperswithcode.com/task/fine-grained-image-classification
>
> [2] Taha, A., Shrivastava, A., & Davis, L.S. (2021). Knowledge Evolution in Neural Networks. ArXiv, abs/2103.05152.
>
> [3] Alabdulmohsin, I.M., Maennel, H., & Keysers, D. (2021). The Impact of Reinitialization on Generalization in Convolutional Neural

---

> > ### Author Response · Authors · 2021-11-13
> > **An initial response to some of the concerns raised in this discussion**
> >
> > We are delighted that both reviewers agree on the potential immense value this perspective can have on deep learning research in general. We thank Reviewer SWHW for engaging in this discussion, and we want to add our thoughts on their comment in this thread. Reviewer MVhg also raised a fascinating set of follow-up directions in their original review, and we will share some thoughts on them in a separate comment. We will also address all reviewers in separate comments.
> >
> > > To clarify, I do not doubt the validity of their experimental setup, but I would argue that solving image classification on extremely small datasets without transfer learning is a niche task with limited significance. Furthermore I'm afraid that not emphasizing this in the paper runs a risk of misleading some readers into thinking that the experimental setup is representative of a typical image classification task.
> >
> > The core contribution of this work lies in proposing the forget-and-relearn hypothesis, which is a new perspective on the iterative training of neural networks. We gather a number of existing iterative algorithms in the literature and study them under the light of this new framework. We introduce LLF and PBF to showcase that existing work can be readily improved in _their respective settings_ by using more targeted forgetting methods, thus demonstrating the explanatory power and practical utility of this perspective. Although our hypothesis is stated in general terms, we do not claim that the specific algorithms we study here should apply in _all settings_. Though some form of LLF and PBF _may_ be more widely beneficial, this is yet to be shown and we view this as outside of the scope of the current paper. We will make this more clear in the final draft and highlight the limitations that Reviewer SWHW pointed out.
> >
> >
> > >I would argue that solving image classification on extremely small datasets without transfer learning is a niche task with limited significance.
> >
> > > For what I can tell, LLF outperforms one other forgetting-based technique on a number of small image classification tasks. To the best of my (limited) knowledge, the comparison is done in a not-very-competitive setting, as most of these datasets can be solved with significantly higher accuracy. For instance, based on [1], both CUB and Stanford Dogs can be solved with over 90% accuracy with transfer learning based methods.
> >
> >
> > Taha et al. was accepted as an oral at CVPR, which suggests that the community has some interest even in the specific small-data/no transfer scenario they focus on. Indeed, Taha et al. claimed SotA results in this setting. LLF was introduced in the context of improving upon KE. We closely followed their evaluation setup for image classification, with the exception of one ImageNet experiment which showed minor improvements and was not the focus of their study. Nonetheless, we can appreciate the desire to see a broader set of experimental results on LLF and will aim to produce more during the rebuttal period, and hope that it will alleviate some of Reviewer SWHW’s concerns.
> >
> > Alabdulmohsin et al. is a concurrent work that systematically evaluated a number of reinitialization methods on a large number of datasets. The coverage of their evaluation is impressive, but it constitutes the core contribution of their paper. Their evaluation setting is also unrealistic. For example, on the CUB200 dataset, their baseline ResNet50 model achieved a test accuracy of 8.5%, far below the ~70% accuracy that our models obtain. In comparison, our contribution extends far beyond LLF. We propose a new perspective on iterative training, study language emergence and compositionality, and correct a number of misconceptions in prior work. These are in addition to our LLF results which significantly outperform both KE and Alabdulmohsin et al.
> >
> >
> > > I believe that the current experiments are too weak to support a major claim about how forgetting can improve DNNs in general
> >
> > As reviewer MVhg suggested, the value of our work can be likened to that of the lottery ticket hypothesis. The first LTH paper stated the general hypothesis: “dense, randomly-initialized, feed-forward networks contain subnetworks (winning tickets) that—when trained in isolation— reach test accuracy comparable to the original network in a similar number of iterations.” They demonstrated results on a small number of model architectures on MNIST and CIFAR10. A large number of follow up papers have since studied this hypothesis in other settings (larger datasets, NLP, RL, etc). As Reviewer SWHW pointed out, studying how forget-and-relearn can be leveraged in new settings will surely be a focus of future research in this direction.
> >
> > We would love to hear your thoughts and would welcome any discussion.

---

> > ### Comment · Reviewer_MVhg · 2021-11-23
> > **To me this is really a 10**
> >
> > This work is not the new state-of-the-art in image classification (see https://arxiv.org/pdf/2003.14415.pdf, Figure 2).
> >
> > I believe that the goal of such kinds of papers is to furnish enough evidence and understanding of what is happening under the hood. So that research engineers consider it valuable to test the new method on very big datasets such as ImageNet, which today are already a sort of production phase, given the computational cost and the low-level tricks needed to train a NN on them, which are prohibitively expensive especially for academic research.
> >
> > In my opinion, this work is doing an excellent job in providing evidence and furnishing this understanding, I think this is very good research.
> >
> > Clearly, [2] and [3] are linked to the methods proposed, but this work is more conceptual and makes the effort of distilling a clear hypothesis. I think this work is what a reader interested in the topic would search after having listened to the CVPR oral of [2].
> >
> > Moreover, I should notice that [3] is not yet published.

---

> ### Author Response · Authors · 2021-11-20
> **Response to review**
>
> Thank you for your time and effort in reviewing our paper. We are very grateful for your vote of confidence in our work. We feel excited about the many future extensions of the ideas in this paper, including those in reinforcement learning, continual learning, and a better understanding of LLF as you point out. We also wanted to share a comment on two of the questions you list below.
>
> > The accuracy gains persist with larger datasets such as Imagenet? How many forget and relearn cycles we should perform? Can we simply wait for convergence also for big datasets?
>
> These are indeed important practical questions. We think that the performance gains of particular algorithms like LLF will vary by dataset, and will be influenced by the interplay between factors such as dataset complexity, model architecture, and model capacity. Future work should also involve designing more sophisticated forgetting operations that better account for this interaction.
>
> > Do we register further gains in accuracy by gradually forgetting fewer layers in LLF?
>
> Intuitively, this seems reasonable. However, we ran a few experiments to test this and it appears not to be the case so far, at least in the settings we tested. This is also similar to the layer-wise (LW) proposal we compared to in our paper, which showed inferior performance. We speculate that the optimal layer may be more influenced by characteristics of the learning task itself, and less by where we are in the training trajectory.
>
> Thank you again for your time and for suggesting these thought-provoking future directions! We would be happy to have further discussions or answer any other questions you may have.

---

### Official Review · Reviewer_SWHW · 2021-11-03

**Correctness:** 2
**Technical Novelty And Significance:** 3
**Empirical Novelty And Significance:** 2
**Recommendation:** 6
**Confidence:** 4

**Main Review:**

This paper proposes a curious new perspective on forgetting that can have significant potential impact on the deep learning research in general. Since forget-and-relearn is a general framework, it can improve a broad range of deep learning applications: not only low-resource image classification and language emergence, but a myriad of others including more practical vision tasks, nlp tasks, speech processing, RL applications and untold others. However, there are significant issues in the current version of the paper that prevent me from recommending acceptance.

The first and main issue is that claiming forget-and-relearn as a general framework **needs more experimental support**. For each major conclusion (difficult examples, compositionality), it is important to check that it holds across many setups with different dataset sizes, training objectives and model architectures. However, this paper only evaluates them on two relatively niche problems. In contrast, related works such as Taha et al. (2021) make a much weaker claims, concerning only with small datasets in computer vision, and yet they explore a more diverse set of tasks including ImageNet classification.

My second (and less significant) concern is that authors do not compare how forget-and-relearn relates to prior art that also connects forgetting and improved generalization, albeit using different terms. One potentially relevant work (R Schwartz-Ziv and N. Tishby, 2017) studies training neural networks from the information perspective and finds that there is a distinct phase where neural network improves generalization by reducing the amount of information about input features in its intermediate representations. Another is Furlanello et al. (2018) where authors propose born-again networks - a form of knowledge distillation where multiple copies of the same network are repeatedly re-trained from scratch to improve the training performance. However, unlike forget-and-relearn where model retains some of its weights, born-again networks retains the useful information from the previous versions using knowledge distillation.

My final and least important issue is the presentation. While the paper is generally well-written, it has a rather unconventional structure with mixing theoretical and experimental sections. While there deviating from the conventional structure *can* sometimes be justified, it makes the paper more difficult to follow. In the case of this specific paper, I do not understand the motivation for changing the structure, especially since many related papers (Taha et al. 2021, Alabdulmohsin et al 2021) managed without it.

References:
- Taha, A., Shrivastava, A., & Davis, L.S. (2021). Knowledge Evolution in Neural Networks. ArXiv, abs/2103.05152.
- Alabdulmohsin, I.M., Maennel, H., & Keysers, D. (2021). The Impact of Reinitialization on Generalization in Convolutional Neural Networks. ArXiv, abs/2109.00267.
- Shwartz-Ziv, R., & Tishby, N. (2017). Opening the Black Box of Deep Neural Networks via Information. ArXiv, abs/1703.00810.
- Furlanello, T., Lipton, Z.C., Tschannen, M., Itti, L., & Anandkumar, A. (2018). Born Again Neural Networks. ICML.




**Summary Of The Paper:**

The paper attempts to unify several prior observations on how partial forgetting during neural network training can affect the final training results. Authors propose forget-and-relearn, a general framework that explains how partial (but not total) one or several times midway through training can improve the final result by mostly forgetting "undesirable" information. To test their hypothesis, authors run experiments in image classification and language emergence, where they partially "forget" learned parameters during training using either later-layer forgetting (LLF) or partial balanced forgetting (PBF), and measure how forgetting affects overall model performance and some specific behaviors. Authors show that a simple perturbations such as "forgetting" layers later in the network can improve performance by disproportionally forgetting difficult and mislabeled examples. In the language emergence task, authors show how forget-and-relearn improves compositionality of the learned language by disproportionally forgetting the non-compositional features.

**Summary Of The Review:**

The paper opens a new and potentially immensely useful perspective on training deep neural networks. However, making claims this strong requires proportionally strong and versatile supoort of experiments and/or theoretical results. Therefore, while this paper has promise, I find its current form too raw and unpolished for publication in a major venue such as ICLR.

---

> ### Author Response · Authors · 2021-11-19
> **Response to review [2/2]**
>
> > this paper only evaluates them on two relatively niche problems.
>
> The two tasks we study, namely image classification and language emergence, are very different problems. Thus, although these demonstrations are limited in scope, we think they offer compelling evidence that forget-and-relearn isn’t an isolated phenomenon.
>
> > My second (and less significant) concern is that authors do not compare how forget-and-relearn relates to prior art that also connects forgetting and improved generalization, albeit using different terms.
>
> Thank you for drawing these connections. We have added a more thorough discussion about other forms of forgetting in Section A5, where we discuss born-again networks, Schwartz-Ziv & Tishby, 2017, and several other works.
>
> > My final and least important issue is the presentation. While the paper is generally well-written, it has a rather unconventional structure with mixing theoretical and experimental sections. While there deviating from the conventional structure can sometimes be justified, it makes the paper more difficult to follow. In the case of this specific paper, I do not understand the motivation for changing the structure, especially since many related papers (Taha et al. 2021, Alabdulmohsin et al 2021) managed without it.
>
> The paper indeed deviates from traditional structure. As the paper covers a broad range of topics, including motivating and stating the hypothesis, explaining prior works in two very different domains, and having multiple experimental sections with different purposes, we felt that the conventional structure would be too inefficient. If there are any specific points of difficulty, we hope to address it.
>
>
> Thank you again for your constructive feedback! We hope that we were able to address your concerns regarding this paper, and we would welcome any further discussion if not. We also hope you may consider raising the score if your concerns are alleviated.

---

> > ### Comment · Reviewer_SWHW · 2021-11-30
> > **Short follow-up**
> >
> > After reading the author's response (both to my review and others) and the revisions, I'm raising my score to 6.
> >
> > For what it's worth, I also apologize for the late response.

---

> ### Author Response · Authors · 2021-11-19
> **Response to review [1/2]**
>
> Thank you for the thoughtful and constructive review of our paper. We are happy that you share our enthusiasm about the significant potential impact this perspective can have on training deep neural networks in general. We’d also like to address the weaknesses you’ve pointed out and share some new experimental results.
>
> > claiming forget-and-relearn as a general framework needs more experimental support
>
> We generally agree with your sentiment. We think this could also be seen as a positive, as it points to the large amount of interesting questions that this hypothesis opens up. We pose forget-and-relearn as a general hypothesis in order to suggest a shift in thinking from “learning desirable information” to “forgetting undesirable information”. How best to do so in different problems and settings is a deep question, which we think is better captured by an entire line of work rather than a single paper.
>
> We also want to clarify that the proposed LLF and PBF are specific examples of forget-and-relearn, and should not be equated with the general hypothesis. In the paper, we show that many existing algorithms can be unified under this framework and better understood through the disproportionate forgetting of undesirable information. To showcase the explanatory power of this hypothesis, we make only slight modifications to the forgetting operation, and keep all other aspects of the algorithm and problem setting the same. Just like their original papers, we do not wish to claim that these particular algorithms will improve results in all settings, and we have updated the paper to make this more clear (see Section 4.1, paragraph 7).
>
> > For each major conclusion (difficult examples, compositionality), it is important to check that it holds across many setups with different dataset sizes, training objectives and model architectures.
>
> Thank you for raising the concern about broadening the dataset and model architectures we consider in our existing experiments. We have run new experiments to address these concerns.
>
> We extended our analysis of weight perturbation on difficult examples to CIFAR-10 on a 4-layer convolutional model (Conv4) and a ResNet18 model, and to ImageNet on ResNet50. Detailed results and discussion can be found in Section A1.2. We found reliable trends for both KE-style and IMP-style forgetting for all datasets.
>
> Below we include a quick summary of KE-style forgetting, which reinitializes a random subset of the network’s weights. We report easy / hard example accuracy post forgetting, with example difficulty defined in terms of output margin (as done in the paper).
>
> |                                        | Conv4 Easy / Hard Acc (CIFAR-10) | ResNet18 Easy / Hard Acc (CIFAR-10) | ResNet50 Easy / Hard Acc (ImageNet)|
> | -------------------------- | -------------------------------- | ------------------------------------- | ---------------------------------- |
> | KE-Style Forgetting      | 61.6% ( $\pm$7.1%) / 41.0% ( $\pm$ 14.7%) | 78.7% ($\pm$ 7.8%) / 53.0% ($\pm$ 15.7%) | 45.6% ($\pm$ 0.7%) / 17.9% ($\pm$ 0.3%) |
>
> We further observe that when compared to the standard IMP-style forgetting, IMP-forgetting with late resetting results in a significantly larger difference between the post-forgetting accuracy of easy and hard examples. It is well demonstrated that late resetting (i.e. rewinding kept weights to early stage of training rather than to initial values) is necessary to achieve good performance, especially in larger models and datasets [1]. Our results demonstrate that IMP with late resetting achieves more targeted forgetting as the hard example group is much more severely affected than the easy example group, which allows it to better leverage the forget-and-relearn process under our hypothesis. This offers another account for why late resetting results in better performance, further showcasing the explanatory power of this perspective.
>
> For LLF, we ran experiments using ResNet50 on Tiny-ImageNet. As shown in Table A7, we observe an improvement in performance, with the test accuracy going from 54.4% to 56.9%.
>
> We are also running experiments on CIFAR-10 and CIFAR-100 on competitive baselines from the literature. These results are currently pending, though we do not expect significant gains in this setting, which is in line with Taha et al. We will include these results in the final version for completeness. Computational constraints prevented us from running ImageNet experiments.
>
>
> [1] https://arxiv.org/abs/1903.01611

---

### Official Review · Reviewer_UMJe · 2021-11-03

**Correctness:** 3
**Technical Novelty And Significance:** 2
**Empirical Novelty And Significance:** 2
**Recommendation:** 6
**Confidence:** 4

**Main Review:**

# Update after rebuttal
* CF was just an example, main point is to provide a better judgment for readers, hence it is necessary to go beyond accuracy and report various metrics. An important point to highlight, given current work is highly empirical, is to show why this method works rather than showing it works.
* Authors should report other metrics such as Precision/Recall, AUC.
* Authors should avoid using any strong claims/words without having any theoretical evidence.
* I agree work is interesting and offers a unique direction for training DNNs, but more work is needed to back a few claims presented in this work.
* Visualizing filters at later layers will further strengthen this work.
* Good job in modifying the work and providing additional results.
* Overall authors have addressed major concerns and confusion.

# Strength
* Easy to understand the paper, well written
* Good sets of experiments and sufficient information provided to replicate the results.

# Weakness
* Strong claims and experiments fail to justify those.
* Comparison with human or brain should be avoided, in fact, recent findings suggest that forgetting is also an essential part, and citations to such works are essential to motivate this research (https://www.nature.com/articles/d41586-019-02211-5).
* Accuracy is not an efficient metric to analyze forgetting in neural networks.
Suppose the hypothesis shows that forgetting and relearning is essential. In that case, the author should conduct experiments showing backward and forward transfers and analyze how much information you lose in the forgetting phase. Show fisher information matrix before, after forgetting, and after relearning, and answer if the same weights are active during all three stages.

* I am also not convinced of the way experiments are conducted; why only tune the last few layers? This is a kind of transfer learning; you keep representation intact and tune on newer tasks or prune on the current task, allowing the least active neurons to change weights to generalize to a given task. It is pruning with sparsity added to obtain low-layer features.  Adding sparsity always helps; even in continual learning, it has shown advantages, which current work fails to cite [3-4]. So I won’t call this forgetting which is misleading, but another form of sparse pruning approach should be called reinitialize and pruning.

* One can achieve sparsity by even using regularization [1], or prune weights using activation functions such as neural activity sparsity [2] or k-winner take all (kWTA)[5]; authors should at least compare with these approaches and show how efficient and computationally efficient their pruning strategy is. Hence I am not convinced by the results, given that important baselines are missing.

* Co-adaption will always lead to overfitting on later layers; this is not true. It may be true in some scenarios, but not for all. I would avoid such claims without experiments or keep language-neutral, saying other effects can also be seen for various models.

* Large scale experiments should be performed with natural images; how does this affect hold for cifar and imagenet. MNIST, there are very few difficult examples; even those examples share most of the representation with other not-so-difficult examples. Another approach would be augmenting data and creating examples that are difficult to understand for neural networks and showing how this pruning-based approach would work and scale.

* In terms of comparison with LW, you should also report performance on their benchmarks. Additionally, the hyperparameter introduced in their work is essential, and performance can vary if you do not carefully tune it. Did you tune it? If not, reporting numbers on their benchmark would be handy. I would like to know the computational cost introduced for all phases (training, forget and learn) and memory footprint (given you copy the network).



* [1] https://pubmed.ncbi.nlm.nih.gov/25987366/
* [2] https://arxiv.org/abs/2104.06153
* [3] https://arxiv.org/abs/1801.01423
* [4] https://arxiv.org/abs/1905.10696
* [5] https://proceedings.neurips.cc/paper/2015/file/5129a5ddcd0dcd755232baa04c231698-Paper.pdf

**Summary Of The Paper:**

# Interesting Idea, but needs major revision and additional results
In this work, the authors introduced forgot and relearn framework to unify disparate existing iterative algorithms. They draw several insights and show that iterative training can achieve stable performance and generalize better on various tasks. They propose various hypotheses showing how forgetting undesired information can help calculate the importance of features important for downstream tasks. These back these claims by conducting experiments on various benchmarks.

**Summary Of The Review:**

# Additional experiments and visualizations is necessary
* Besides the points stated above, I am also interested in seeing the statistical significance of the result. Can the author report a standard error for all experiments for k trials (at least k >2)? Please show hidden representation with later layers during the before phase, forget/reinitialize phase and relearn/pruning phase.
* Additionally, the effect of sparsity on model performance, how to ensure signals or later features are sparse enough for the model to generalize better. Having theory to back these claims is ideal, given current work is heavily experimental, I would encourage authors to add more insight and results to back their claims.

---

> ### Author Response · Authors · 2021-11-16
> **Response to review [3/3]**
>
> > Comparison with human or brain should be avoided, in fact, recent findings suggest that forgetting is also an essential part, and citations to such works are essential to motivate this research (https://www.nature.com/articles/d41586-019-02211-5).
>
> We only draw connections to human forgetting in the introduction as a way to motivate the idea that forgetting can be useful, as you also point out. Thank you for the reference, we will include this in the text.
>
> > Co-adaption will always lead to overfitting on later layers; this is not true. It may be true in some scenarios, but not for all. I would avoid such claims without experiments or keep language-neutral, saying other effects can also be seen for various models.
>
> We do not wish to claim this statement to be true in the paper. This is only offered as a potential hypothesis for why iterative retraining with LLF could help generalization in the small-data setting that we study. In fact, we show in Section 5.1 that this is _not_ a valid reason for the improved generalization. We will make this more clear in the final manuscript.
>
> > In terms of comparison with LW, you should also report performance on their benchmarks. Additionally, the hyperparameter introduced in their work is essential, and performance can vary if you do not carefully tune it. Did you tune it? If not, reporting numbers on their benchmark would be handy. I would like to know the computational cost introduced for all phases (training, forget and learn) and memory footprint (given you copy the network).
>
> Unlike LLF, LW does not have any additional hyperparameters in its canonical form (i.e. N=1). This is the form that LW authors use in their reported results, and we use the same setting in our comparison.
>
> We do not copy the network in LLF, we simply reinitialize later layers and allow the same model to continue to train. Compared to a single generation of training, LLF trained for T generations will have T times the computational cost. Compared to our long baselines, LLF has the same computational cost barring a small additional cost of weight reinitialization.
>
> > Besides the points stated above, I am also interested in seeing the statistical significance of the result. Can the author report a standard error for all experiments for k trials (at least k >2)?
>
> We have run all our experiments for a minimum of 3 trials. We will report the standard errors for our results in the updated manuscript.
>
> We thank the reviewer once again for their feedback. We hope we were able to address your concerns and would welcome any further discussion if not. We also hope you can consider raising your score if your concerns were sufficiently addressed.

---

> ### Author Response · Authors · 2021-11-16
> **Response to review [2/3]**
>
> > Accuracy is not an efficient metric to analyze forgetting in neural networks. Suppose the hypothesis shows that forgetting and relearning is essential. In that case, the author should conduct experiments showing backward and forward transfers and analyze how much information you lose in the forgetting phase. Show fisher information matrix before, after forgetting, and after relearning, and answer if the same weights are active during all three stages.
>
> Our notion of “forgetting” differs from that of catastrophic forgetting and serves a different purpose. In the algorithms we study, relearning is always done on the original task, thus we do not think backward and forward transfers or Fisher information matrix is necessary here.
>
> > I am also not convinced of the way experiments are conducted; why only tune the last few layers? This is a kind of transfer learning; you keep representation intact and tune on newer tasks or prune on the current task, allowing the least active neurons to change weights to generalize to a given task. It is pruning with sparsity added to obtain low-layer features. Adding sparsity always helps; even in continual learning, it has shown advantages, which current work fails to cite [3-4]. So I won’t call this forgetting which is misleading, but another form of sparse pruning approach should be called reinitialize and pruning.
>
> > One can achieve sparsity by even using regularization [1], or prune weights using activation functions such as neural activity sparsity [2] or k-winner take all (kWTA)[5]; authors should at least compare with these approaches and show how efficient and computationally efficient their pruning strategy is. Hence I am not convinced by the results, given that important baselines are missing.
>
> > Additionally, the effect of sparsity on model performance, how to ensure signals or later features are sparse enough for the model to generalize better. Having theory to back these claims is ideal, given current work is heavily experimental, I would encourage authors to add more insight and results to back their claims.
>
> For later layer forgetting (LLF), we do not just tune the last few layers; we re-initialize the weights of the last few layers after every generation, after which the entire network is retrained. Essentially, this is the same as normal end-to-end training but with weights of the last few layers being re-initialized at regular intervals. The simplest form of forget-and-relearn would look like the following:
>
> `train entire model for M steps -> forget some information -> train entire model for M steps -> forget some information -> ... for T generations `
>
> We are not performing pruning or trying to achieve sparsity directly or indirectly, and we make no such claims in the paper.
>
> The forget-and-relearn hypothesis suggests that by selectively forgetting undesirable information, we can steer the model towards more desirable characteristics through iterative retraining. Based on prior works that relate memorization to difficult examples and overfitting, and observations in Baldock et al. [1] showing how difficult examples are learned in the later layers of the network, we hypothesize that reinitializing later layers will disproportionately remove information related to overfitting, thus motivating LLF.
>
> > Large scale experiments should be performed with natural images; how does this affect hold for cifar and imagenet. MNIST, there are very few difficult examples; even those examples share most of the representation with other not-so-difficult examples. Another approach would be augmenting data and creating examples that are difficult to understand for neural networks and showing how this pruning-based approach would work and scale.
>
> Thank you for your suggestions. We have extended our analysis of difficult examples to CIFAR-10 on a 4-layer convolutional model (Conv4) and a ResNet18 model, and to ImageNet on ResNet50. We will update the paper with full details of these experiments. The summary is that we find reliable trends for both KE-style and IMP-style forgetting for all datasets. Below we include a quick summary of KE-style forgetting, which reinitializes a random subset of the network’s weights. We report easy / hard example accuracy post forgetting, with example difficulty defined in terms of output margin (as done in the paper).
>
> |                                        | Conv4 Easy / Hard Acc (CIFAR-10) | ResNet18 Easy / Hard Acc (CIFAR-10) | ResNet50 Easy / Hard Acc (ImageNet)|
> | -------------------------- | -------------------------------- | ------------------------------------- | ---------------------------------- |
> | KE-Style Forgetting      | 61.6% ( $\pm$7.1%) / 41.0% ( $\pm$ 14.7%) | 78.7% ($\pm$ 7.8%) / 53.0% ($\pm$ 15.7%) | 45.6% ($\pm$ 0.7%) / 17.9% ($\pm$ 0.3%) |
>
> As we can see, these forms of weight perturbation more severely affect difficult examples across datasets and model architectures.
>
> [1] https://arxiv.org/abs/2106.09647

---

> ### Author Response · Authors · 2021-11-16
> **Response to review [1/3]**
>
> Thank you for your time and effort in writing this review.
>
> We think that there may be a misunderstanding of our paper, as we could not relate some of the points raised in the review to our presented work. We will attempt to clarify these misunderstandings in our response. However, if we did not understand your questions as intended, please let us know and we will offer further clarifications.
>
> In this work, we introduce a new notion of “forgetting” for training neural networks, which is separate from the existing notion of “catastrophic forgetting” in the continual learning literature. Forgetting in our paper is a general notion that describes _any_ action resulting in reduced training performance (thus enabling relearning) while retaining partial knowledge (thus not having to restart from scratch). We do not study continual learning or transfer learning settings in this paper, though it may be fruitful to design forget-and-relearn algorithms for these settings in future work.
>
> We introduce the concept of forget-and-relearn as a framework for understanding and designing iterative training algorithms. Under this perspective, a key factor to the success of iterative algorithms is a forgetting operation that disproportionately removes undesirable information. We recast a number of existing iterative algorithms in the literature as instances of forget-and-relearn, and highlight what their forgetting operation is. We then show that these forgetting operations disproportionately affect undesirable information in their respective settings. This is the first step to supporting the forget-and-relearn interpretation of existing algorithms. We then show that by designing more targeted forgetting operations (LLF and PBF), we can significantly improve the performance of existing algorithms, which further supports the explanatory power and utility of the forget-and-relearn hypothesis.
>
> We will provide a point-by-point response in the next comment.

---

### Author Response · Authors · 2021-11-23
**Thank you for your reviews; new revision submitted; summary of changes below**

We are very grateful for all the valuable suggestions from the reviewers. We have made several improvements to the paper, including clarifications, new discussions, and new results. Here is a summary of these changes:

1. Added partial weight perturbation analysis on CIFAR-10 and ImageNet in Appendix A1.2, indicating that KE-style and IMP-style forgetting disproportionately affect undesirable information in these datasets as well. We also discover a possible explanation for why “late resetting” is required when scaling up lottery ticket experiments, further demonstrating the power of the forget-and-relearn perspective.

2. Added later layer forgetting experiments for Tiny-ImageNet, CIFAR-10, and CIFAR-100 in Appendix A2.3. We find significant gains for Tiny-ImageNet, but not for CIFAR-10 and CIFAR-100.

3. Added a new paragraph in Section 4.1 to clarify that we do not present LLF as a general algorithm that works across all settings, but instead as an instance of forget-and-relearn aimed at directly improving upon KE in the small data setting.

4. Added a new discussion relating our notion of forgetting to other ones in Appendix A5 and new references in Section 1.

5. Added more details regarding hyperparameter selection in Appendix A2.1.

6. Added standard errors for our results in Table 1.

7. Clarified that we do not claim that co-adaptation will always lead to overfitting.

8. Added a new Nature reference in Section 1.

We thank the reviewers once again for their time and effort. We hope we have been able to address your concerns. We are happy to continue discussing our paper.

---

### Author Response · Authors · 2021-11-29
**Summary of changes**

Dear reviewers / AC,

Thank you all again for your time and valuable comments. With one day left in the rebuttal discussion, we are eager to hear from the reviewers about their thoughts on our responses. In particular, we hope to hear from reviewers UMJe, SWHW, and hRdL, because our responses contained important clarifications, and we have made changes to the paper based on these reviews.

A quick summary of draft updates below:

- Based on reviewer UMJe's comments, we have updated the language in several instances, we have also added new experimental results and standard errors.

- Based on reviewer SWHW's comments, we have added new experimental results on larger datasets and more models, which increase the completeness of our evaluations and have revealed new insights into iterative magnitude pruning. We also added more thorough discussions on how our work relates to other forms of forgetting in the literature.

- Based on reviewer hRdL's comments, we have included more dataset and experimental details, and discussed guidance for how to apply forget-and-relearn in the general case.

We would appreciate it if reviewers UMJe, SWHW, and hRdL could kindly take a look at the revision and our response to your comments to see if your concerns are sufficiently resolved and if your evaluation is hopefully more positive. Thank you again for your valuable time.

Best regards,

Authors

---

### Decision · Program_Chairs · 2022-01-20

**Decision:**

Accept (Poster)

**Comment:**

In the paper, it introduces a forget-and-relearn framework to the iterative learning algorithm. It provides serval new insight that forgetting could be favorable to learning and validates the insights via image classification and language tasks. The idea is novel and inspiring. Although there are some debates on the experiment and the generality of the proposed method, I think authors answered those questions decently and many researchers would be interested in this direction.